# Treponema pallidum subsp. pallidum with an Artificially impaired TprK antigenic variation system is attenuated in the Rabbit model of syphilis

Emily Romeis[1], Nicole A. P. Lieberman[2], Barbara Molini[1], Lauren C. Tantalo[1], Benjamin Chung[2], Quynh Phung[2], Carlos Avendaño[2], Anastassia Vorobieva[3,4], Alexander L. Greninger[2,5], Lorenzo Giacani[1,6]*

1 Department of Medicine, Division of Allergy and Infectious Diseases, University of Washington, Seattle, Washington, United States of America, 2 Department of Laboratory Medicine and Pathology, University of Washington, Seattle, Washington, United States of America, 3 VIB-VUB Center for Structural Biology, VIB, Brussels, Belgium, 4 Structural Biology Brussels, Vrije Universiteit Brussel, Brussels, Belgium, 5 Vaccine and Infectious Disease Division, Fred Hutchinson Cancer Research Center, Seattle, Washington, United States of America, 6 Department of Global Health, University of Washington, Seattle, Washington, United States of America

* giacal@u.washington.edu

**Data Availability Statement:** All data are available in the manuscript files and Supplementary materials. Whole genome sequencing data are

## Abstract

### Background

The TprK protein of the syphilis agent, *Treponema pallidum* subsp. *pallidum* (*T. pallidum*), undergoes antigenic variation in seven discrete variable (V) regions via non-reciprocal segmental gene conversion. These recombination events transfer information from a repertoire of 53 silent chromosomal donor cassettes (DCs) into the single *tprK* expression site to continually generate TprK variants. Several lines of research developed over the last two decades support the theory that this mechanism is central to *T. pallidum*'s ability for immune avoidance and persistence in the host. Structural and modeling data, for example, identify TprK as an integral outer membrane porin with the V regions exposed on the pathogen's surface. Furthermore, infection-induced antibodies preferentially target the V regions rather than the predicted β-barrel scaffolding, and sequence variation abrogates the binding of antibodies elicited by antigenically different V regions. Here, we engineered a *T. pallidum* strain to impair its ability to vary TprK and assessed its virulence in the rabbit model of syphilis.

### Principal findings

A suicide vector was transformed into the wild-type (WT) SS14 *T. pallidum* isolate to eliminate 96% of its *tprK* DCs. The resulting SS14-DC^KO strain exhibited an *in vitro* growth rate identical to the untransformed strain, supporting that the elimination of the DCs did not affect strain viability in absence of immune pressure. In rabbits injected intradermally with the SS14-DC^KO strain, generation of new TprK sequences was impaired, and the animals developed attenuated lesions with a significantly reduced treponemal burden compared to

available on GenBank under BioProject PRJNA909291. An interactive version of the heatmaps reported in the manuscript (Fig 6) and Supplementary Material (S1–S6 Figs.) is provided at at https://github.com/greninger-lab/Impaired-TprK-Antigenic-Variation.

**Funding:** This work was supported by NIAID grant number U19AI144133 (Project 2 and Genomics and Isolation Core; Project 2 leader: L.G.; Core leaders: L.G. and A.L.G.; PI: Anna Wald, University of Washington) and by Open Philanthropy pledge # 8394150 (to L.G.). The funders had no role in study design, data collection, and analysis, decision to publish, or preparation of the manuscript.

**Competing interests:** The authors have declared that no competing interests exist.

control animals. During infection, clearance of V region variants originally in the inoculum mirrored the generation of antibodies to these variants, although no new variants were generated in the SS14-DC$^{KO}$ strain to overcome immune pressure. Naïve rabbits that received lymph node extracts from animals infected with the SS14-DC$^{KO}$ strain remained uninfected.

## Conclusion

These data further support the critical role of TprK in *T. pallidum* virulence and persistence during infection.

## Author summary

Syphilis is still endemic in low- and middle-income countries, and it has been resurgent in high-income nations, including the U.S., for years. In endemic areas, there is still significant morbidity and mortality associated with this disease, particularly when its causative agent, the spirochete *Treponema pallidum* subsp. *pallidum* (*T. pallidum*) infects the fetus during pregnancy. Improving our understanding of syphilis pathogenesis and *T. pallidum* biology could help investigators devise better control strategies for this serious infection. Now that tools to genetically manipulate this pathogen are available, we can engineer *T. pallidum* strains lacking specific genes or genomic regions known (or believed) to be associated with virulence. This approach can shed light on the role of the ablated genes or sequences in disease development using loss-of-function strains. Here, we derived a knockout (KO) *T. pallidum* mutant (SS14-DC$^{KO}$) impaired in its ability to undergo antigenic variation of TprK, a protein that has long been hypothesized to be central in evasion of the host immune response and pathogen persistence during infection. When compared to the WT isolate, which is still capable of antigenic variation, the SS14-DC$^{KO}$ strain is significantly attenuated in its ability to proliferate and to induce early disease manifestations in infected rabbits. Our results further support the importance of TprK antigenic variation in syphilis pathogenesis and pathogen persistence.

## Introduction

Syphilis is a chronic sexually transmitted infection that still represents a concern for public health as it causes significant morbidity and mortality worldwide. The World Health Organization (WHO) estimated that syphilis global incidence is between 5.6 to 11 million new cases every year, while global prevalence ranges between 18 to 36 million cases [1,2]. Although most cases occur in low- and middle-income countries where the disease is endemic, syphilis rates have been on the rise for years in high-income nations, primarily among men who have sex with men (MSM), even though the gap in incidence between MSM and the heterosexual population is gradually reducing [3–8]. In the US, the rate of early syphilis in 2020 (12.7 cases per 100,000 population) represented a 504% increase compared to the rate reported in 2000 (2.1 cases per 100,000 population) [3]. Furthermore, about 1.4 million pregnant women a year acquire syphilis worldwide. Congenital syphilis (CS), resulting from the ability of *Treponema pallidum* subsp. *pallidum* (*T. pallidum*) to colonize and cross the placenta, causes an estimated 305,000 annual cases of fetal loss or stillbirth and 215,000 infants born prematurely and/or with clinical signs of infection [9–11].

In untreated subjects, primary and secondary syphilis generally manifests in distinct clinical stages, followed by a period of asymptomatic latency that, in most cases, will last for the patient's lifetime [12]. Data preceding the introduction of penicillin for syphilis treatment showed that in about 30% of patients, the disease would reactivate after years to decades from the onset of latency and progress to its tertiary stage and its most serious manifestations [13]. Key to this decades-long persistence is the pathogen's ability to evade the robust host immune response that develops during infection. This central feature of *T. pallidum* virulence has long been attributed to antigenic variation of the TprK (Tp0897) protein [12]. This putative integral outer-membrane porin [14] harbors seven discrete variable regions (V1-V7), each predicted by AlphaFold2 (https://AlphaFold.ebi.ac.uk/) to form an external loop at the host-pathogen interface (Fig 1). Sequence variability within the V regions is generated through non-reciprocal segmental gene conversion, a process that transfers information contained in 53 silent

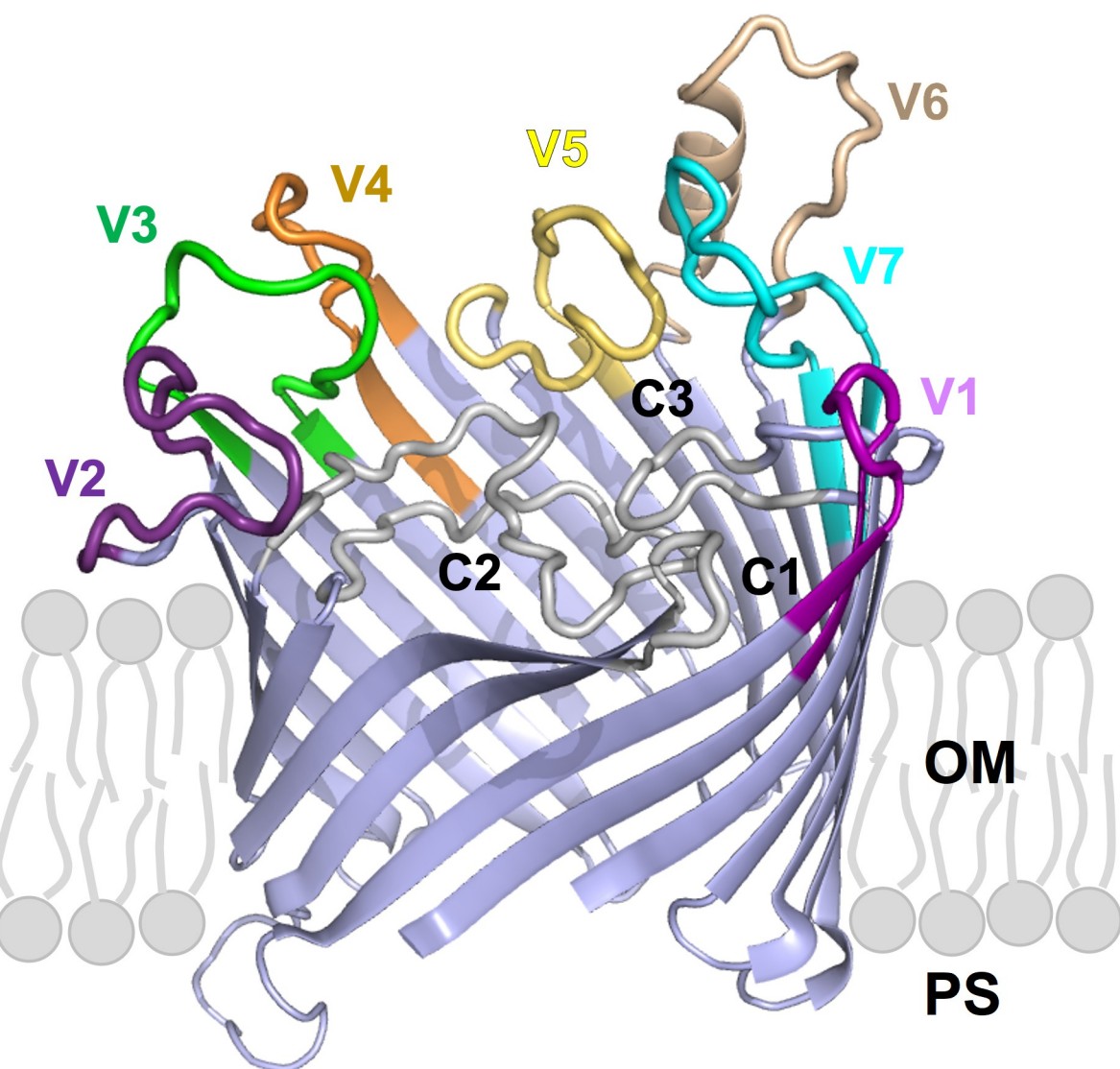

**Fig 1. TprK (sequence AIP85979.1) is predicted by AlphaFold2 to fold into a 20-strands beta-barrel structures with 10 surface-exposed loops.** Variable loops (V1-V7) are highlighted in colour. Conserved loops (C1-C3) are in white, and β-scaffolding are in light blue. OM: outer membrane; PS: periplasmic space.

chromosomal donor cassettes (DCs) into the *tprK* expression site [15,16], located ~175 kbp away from the DCs. Most of the DCs (51/53; 96%) are located downstream of the *tprD* gene (*tp0131*) at nucleotide (nt) position 148,435–152,400, while the remaining two DCs are downstream of the *tp0136* gene at position 159,499–159,576, based on the SS14 strain reference genome sequence (NC_021508.1/CP004011.1) [17]. In a previous study by Addetia *et al.* [17] it was estimated that the total *tprK* diversity that could be generated by the combined usage of all DCs would range between $10^{13}$–$10^{15}$ full-length individual TprK sequences [17].

Aside from sequence variability and potential for generation of quintillions of unique sequences, the role of TprK in immune evasion and pathogen persistence is supported by the evidence that a) sequence diversity accumulates over time during experimental infection [18], b) infection-induced antibodies preferentially target the TprK V regions rather than the predicted β-barrel scaffolding of the protein [19], c) sequence variation abrogates binding of antibodies raised to antigenically different V regions [20], d) TprK variants accumulate more rapidly in immunocompetent hosts compared to immunosuppressed rabbits as the result of immune selection [21,22] and, lastly, e) TprK sequences found in disseminated secondary lesions in the rabbit model differ from those in the original inoculum, supporting that secondary lesions contain treponemes that avoided the initial round of immune clearance associated with the resolution of the primary lesion(s) [23].

Recently, Romeis *et al.* [24] described a procedure that successfully allowed the replacement of a specific *T. pallidum* chromosomal region with a kanamycin resistance ($kan^R$) cassette. This approach uses a suicide vector with the $kan^R$ cloned between homology arms matching the regions upstream and downstream of the sequence to be deleted. This work provided the first experimental evidence that *T. pallidum* could be genetically altered and opened the gateway for systematic derivation of KO mutants to assess their fitness and virulence *in vivo* and shed light on the role of specific genes and or genomic regions in syphilis pathogenesis. Although a *tprK*$^{KO}$ strain could not be derived, to further explore the role of TprK in *T. pallidum* persistence, we ablated the 51 DCs situated downstream of the *tprD* gene in the SS14 *T. pallidum* strain, which resulted in an impaired TprK antigenic variation system. We then evaluated disease progression and the pathogen's ability to proliferate in rabbits infected with the mutant strain (renamed SS14-DC$^{KO}$) in comparison to animals infected with the WT strain, still fully capable of varying TprK.

## Results

### SS14-DC$^{KO}$ mutant treponemes are not impaired in their ability to grow *in vitro* and *in vivo* in a partially immune-privileged site

Treponemes transformed with the p*DC*arms-47p-*kan*$^R$ suicide vector, designed to eliminate the DCs downstream of *tprD* through a double cross-over recombination event, were selected by propagation in selective media (Fig 2A). As a control, the parent SS14 strain was also propagated in both normal and selective media in parallel to the transformed strain. Due to the ~44 hours generation time of the SS14 strain, *T. pallidum* cells needed to be sub-cultured every 14 days until week 13 post-transformation (Passage #8; Fig 2A), and weekly thereafter. Transformed SS14-DC$^{KO}$ treponemes had significantly recovered by 4 weeks post-transformation (Fig 2A) at the time passage #3 was harvested, and approximately 10 treponemal cells per microscopic field (corresponding to ~$10^7$ *T. pallidum* cells/ml) could be counted at this time. The cell density of transformed treponemes increased by week 10 and remained steady with harvests of ~$10^8$ *T. pallidum* cells/ml (Fig 2A). Wild-type parent SS14 cells propagated in selective media could not be seen by dark-field microscopy (DFM) at any time and sub-culturing was stopped at week 8 (Fig 2A). Propagation of the WT parent strain was halted at week 18 (Fig 2A). DNA extracted from treponemes harvested at passage #8 post-transformation was

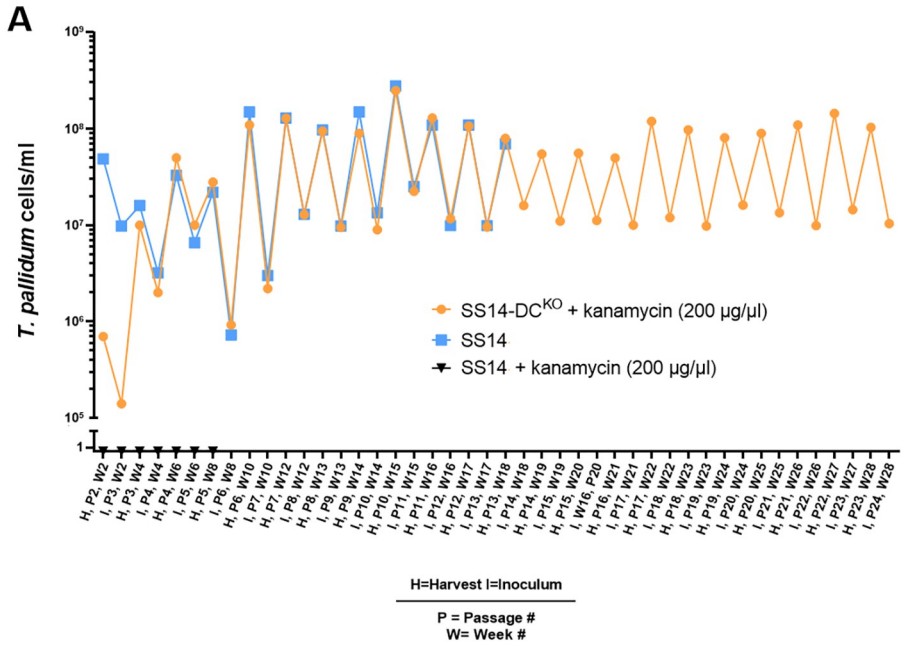

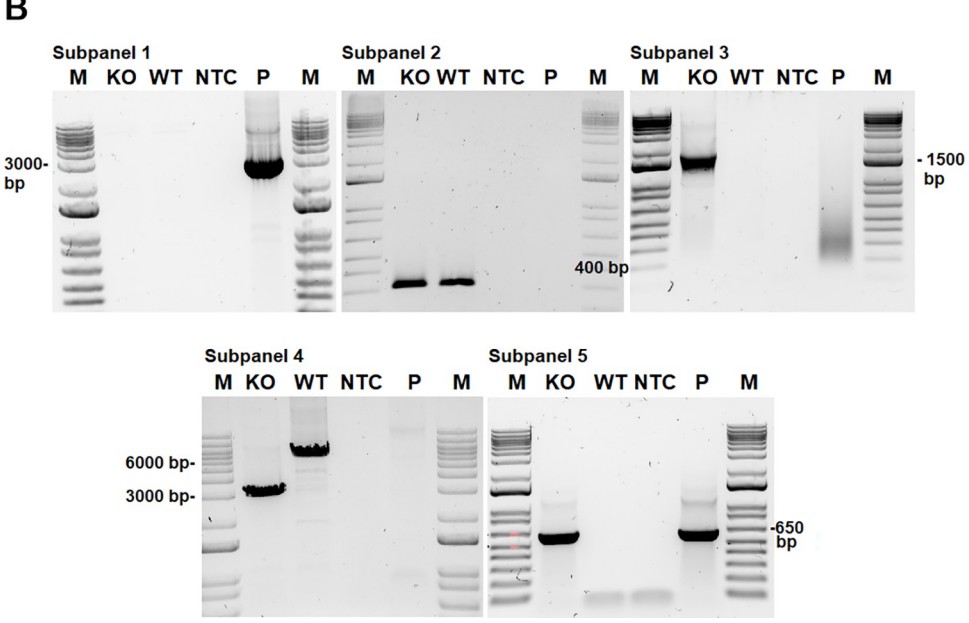

**Fig 2. *In vitro* growth curves of the strains used here and molecular assessment of integration of the *kan*R cassette.** (**A**) Twenty-eight-week *in vitro* growth curve of transformed SS14 *T. pallidum* cells (SS14-DCKO) in selective media (solid orange circles), WT SS14 cells in selective media (solid black triangles), and WT SS14 strain in non-selective media (solid blue squares). Propagation of the parent strain for this experiment stopped at week 18. Parent treponemes propagated in kanamycin-containing media were never countable by DFM. (**B; subpanels 1–5**) Qualitative amplifications to assess *kan*R integration. Primer combinations are reported in Table 1. Subpanels show amplifications a) with vector-specific primers (subpanel #1, 2,967 bp amplicon), yielding an amplicon only with suicide vector control DNA; b) with *tp0574* primers (subpanel #2, 313 bp amplicon), positive in both parent and transformed treponemes; c) with primers annealing to the *kan*R gene and outside of the right homology arm in the *T. pallidum* genome (subpanels #3; 1,846 bp size amplicon), yielding a positive amplicon only with template from transformed treponemes; d) with primers annealing upstream and downstream of the homology arms in the *T. pallidum* genome (subpanel #4, 2,982 bp and 5,980 bp size amplicons), yielding a shorter amplicon size in transformed treponemes due to deletion of the DCs); and primers annealing to the *kan*R gene (subpanel #5, 663 bp size amplicon). M: molecular size marker (bp); KO: SS14-DCKO, WT: parent WT SS14; NTC: No-template control, P: suicide vector (p*DC*arms-47p-*kan*R) plasmid DNA.

**Table 1. Primers used in this study.**

| Targets; application | Forward (F) and reverse (R) primer sequences (5'-3'), and probe (P) if applicable | Amplicon length (bp) | Sub-panel # in Fig 2B |
|---|---|---|---|
| pUC57 vector primers; sequencing, assessing plasmid carry-over | (F) TAAAACGACGGCCAGTGAAT<br>(R) GACCATGATTACGCCAAGC | 2,967 | 1 |
| *tp0574*; qualitative PCR | (F) TGTGGCTCGTCTCATCATGA<br>(R) CTGGGCCACTACCTTCGCAC | 313 | 2 |
| *kan*[R] (F primer), right *DC* flanking region (R primer); qualitative PCR and sequencing (F) | (F) GAGCCATATTCAACGGGAGA<br>(R) CTTCTCCTTCGCCCTCGAC | 1,799 | 3 |
| Left *DC* flanking region (F primer),<br>Right *DC* flanking region (R primer); qualitative PCR | (F) CACAGAAGGGCAGCAGTACA<br>(R) CTTCTCCTTCGCCCTCGAC | 2,982 (SS14-DC[KO]) or 5,980 (WT) | 4 |
| *kan*[R]; qualitative PCR; RT-qPCR | (F) GAGCCATATTCAACGGGAGA<br>(R) ATTCCGACTCGTCCAACATC | 663 | 5 |
| Left *DC* flanking region (F primer),<br>*kan*[R] (R primer); qualitative PCR and sequencing | (F) CACAGAAGGGCAGCAGTACA<br>(R) ATTCCGACTCGTCCAACATC | 1,846 | |
| *tp0131 (tprD)*; RT-qPCR | (F) CACTAGTCTTGGGGACACGC<br>(R) TCCTGGTTGCAATTCACGTA | 273 | |
| *tp0897 (tprK)*; RT-qPCR | (F) GAAAATCGCCTGTGCCCTAC<br>(R) GGTTCCCCACGTTTAGTTAG | 80 | |
| *tp0574*; qPCR, RT-qPCR | (F) CAAGTACGAGGGGAACATCG<br>(R) TGATCGCTGACAAGCTTAGG<br>(P) 6FAM-CGGAGACTCTGATGGATGCTGCAGTT-NFQMGB[1] | 132 | |
| *tp0127*; ddPCR | (F) GCGTGTATGGAGAAGCATTG<br>(R) CGCCTGATAGCCTATATCCAC<br>(P) HEX-AGATATGGT/ZEN/ACTACGCGCCGC-3IABkFQ[2] | 139 | |
| *kan*[R]; ddPCR | (F) CACTCAGGCGCAATCAC<br>(R) CCAGACTTGTTCAACAGGC<br>(P) FAM-ACGGTTTGGTTGATGCGAGTGATTT-BkFQ[3] | 91 | |
| *tp0001 (dnaA)*; ddPCR, | (F) CTCATGGAAATACTGCTCC<br>(R) CGGATACAAAGTTCTCGAAG<br>(P) FAM-AGCTTTCACCCCGACCTGAAC-BkFQ[3] | 135 | |
| *tp0574* promoter; sequencing | (F) AGCGGATCCTCCCAAAAAGA<br>(R) GATTACACCTCCGTATAGAG | N/A | |
| pUC57 vector primers; sequencing, assessing plasmid carry-over | (F) TAAAACGACGGCCAGTGAAT<br>(R) GACCATGATTACGCCAAGC | 2,967 | |

[1]Minor groove binder nonfluorescent quencher

[2]3' Iowa Black fluorescence *quencher*

[3]Black hole fluorescence *quencher*

used for qualitative amplifications (primers in Table 1) designed to confirm integration of the *tp0574* promoter-*kan*[R] sequence where the DC-containing region used to be, as well as lack of residual transformation vector or DNA from the parent strain. Results (Fig 2B) showed that no amplification occurred in treponemal cultures with primers specific for the suicide vector backbone (subpanel #1), confirming lack of residual plasmid DNA. As expected, amplification of the *tp0574* gene was positive for both the parent SS14 and the SS14-DC[KO] strain templates (subpanel #2), confirming the presence of treponemal DNA in both cultures. Furthermore, amplification of the target region with the reverse primer annealing downstream of the right homology arm (hence not cloned into the suicide vector) and a forward primer annealing to the *kan*[R] gene, respectively (subpanel #3), yielded positive amplification only with template from the SS14-DC[KO] strain, confirming that integration of the *tp0574* promoter-*kan*[R] sequence occurred where expected. Amplification with both primers outside of the homology arms yielded amplicons of different sizes (subpanel #4; 2,982 bp for SS14-DC[KO] and 5,980 bp

for the parent strain), confirming deletion of the DCs and lack of parent SS14 DNA in the mutant strain culture. Lastly, amplification with primers specific for the *kan*<sup>R</sup> only amplified the SS14-DC<sup>KO</sup> and the vector control template (subpanel #5).

Rabbit intratesticular (IT) inoculation with either the WT or the SS14-DC<sup>KO</sup> strain was also used to assess whether the mutant strain would be impaired in its ability to grow *in vivo* and induce orchitis. The time to orchitis in WT- and SS14-DC<sup>KO</sup>-infected animals was identical (21 days post-inoculation) and, when harvested, both rabbits yielded comparable treponemal concentrations (~$10^8$/ml). For both animals, testes were minced and extracted in 10 ml of sterile saline.

Altogether, these data confirmed that a) the *kan*<sup>R</sup> gene was integrated into the SS14-DC<sup>KO</sup> strain genome in place of the targeted 51 DCs, that b) no residual suicide plasmid DNA or parent treponemes contaminated the SS14-DC<sup>KO</sup> culture, that c) 96% of the DCs are missing in the mutant strain and lastly, that d) despite this deletion, SS14-DC<sup>KO</sup> treponemes are not impaired in their ability to grow *in vitro* and *in vivo* compared to the WT strain.

## Genome sequencing and droplet digital PCR confirm deletion of 96% of the DCs, absence of additional mutations between the WT and SS14-DC<sup>KO</sup> strains, and similar *tprK* expression level in WT and mutant treponemes

At passage #2–5 (Week 2–6 post-transformation in Fig 2A), ahead of genome sequencing, DNA extracted from SS14-DC<sup>KO</sup> and parent SS14 cells was processed for quantitative droplet digital PCR (ddPCR) to calculate copy number of the *kan*<sup>R</sup> and *tp0127* (contained within the eliminated region containing the DCs) genes, normalized to the copy number of *dnaA* gene (*tp0001*; present in both parent and transformed treponemes). The normalized ratios of either *kan*<sup>R</sup> or *tp0127* were expected to be either zero or ~1.0 depending on the strain analyzed. When the parent SS14 DNA was used as control, the *tp0127*:*dnaA* was near 1.0 over the four analyzed passages (Fig 3A), while the *kan*<sup>R</sup>:*dnaA* ratio was zero due to lack of the *kan*<sup>R</sup> gene in this strain (Fig 3A). In the SS14-DC<sup>KO</sup> strain, the *kan*<sup>R</sup>:*dnaA* ratio neared 1.0 when DNA from passage #3 and #4 was analyzed (Fig 3A), and was 1.0 for passage #5, while the *tp0127*:*dnaA* ratio was zero (due to the absence of *tp0127*), with the exception of passage #2. This result was however expected due to the presence of residual parent SS14 DNA, given that the culture had not been passaged yet after transformation (Fig 3A). These data reiterated that a) integration of the *kan*<sup>R</sup> gene occurred in the DC locus and supported that no extra copies of the *kan*<sup>R</sup> gene existed outside or within the SS14-DC<sup>KO</sup> strain genome. As previously discussed [24], the slightly <1.0 results in some of these assays are likely due to the proximity of the *dnaA* gene (nt position 4–1,395 in the SS14 genome) to the chromosomal origin of replication.

To ascertain that DC deletion did not affect *tprK* transcription, cDNA generated from the SS14-DC<sup>KO</sup> and WT strain RNA, was used to assess transcription of the *tprK* (*tp0897*) and *tp0574* genes (used as housekeeping gene; primers in Table 1). Message quantification (Fig 3B) showed that, like in other *T. pallidum* strains [25], *tprK* is poorly expressed in the SS14 strain compared to *tp0574*, but its expression does not differ significantly between the WT and mutant strains.

Whole genome sequencing (WGS) performed on SS14-DC<sup>KO</sup> treponemes from passage #9 confirmed the absence of reads for the *tprK* DCs in the SS14-DC<sup>KO</sup> strain following assembly to the SS14 reference genome (Fig 3C). Assembly using a reference genome where the DCs were replaced *in silico* with the *kan*<sup>R</sup> cassette (KO reference) was consistent with the replacement of the 51 DCs downstream of *tprD* with the *tp0574* promoter-*kan*<sup>R</sup> sequence (Fig 3D). Because sequencing used enrichment probes specific for multiple WT *T. pallidum* genomes, coverage of the *kan*<sup>R</sup> gene (Fig 3D) was slightly lower than the average compared to the rest of

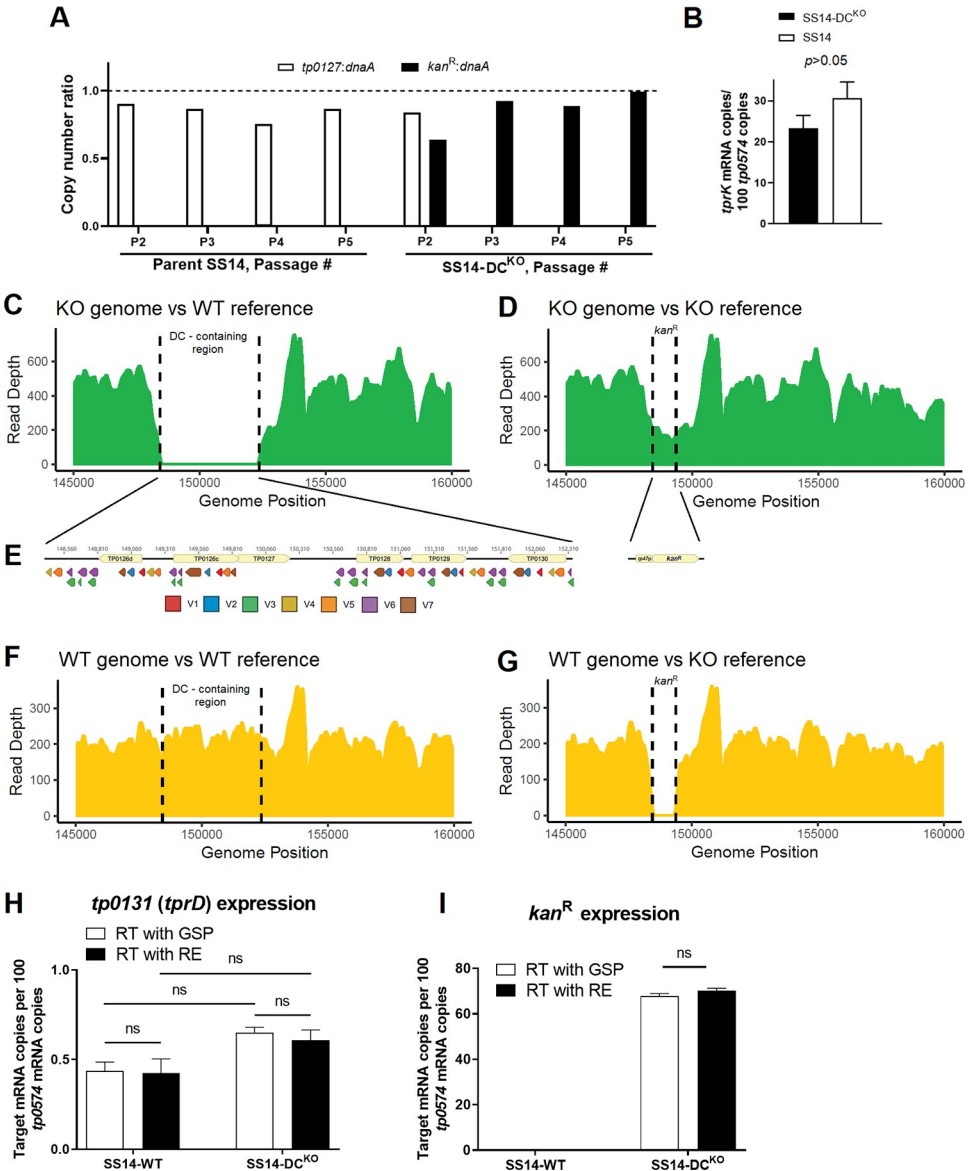

**Fig 3. Confirmation of *kan*^R cassette integration using ddPCR and WGS, and message quantifications. (A)** Ratios between the *kan*^R, *dnaA (tp0001)*, and *tp0127* (contained within the deleted DC) copy number determined by droplet digital PCR (ddPCR) on template DNA from the SS14-DC^KO and parent SS14 strain harvested at passage #2–5 post-transformation. The *kan*^R:*dnaA* ratio for the parent SS14 was zero at each passage. **(B)** Message quantification of *tprK* (normalized to *tp0574*) in SS14-DC^KO and parent SS14 strain propagated *in vitro* showing comparable expression of *tprK* between the control and transformed strain. Significance (*p*<0.05) was assessed by performing Student's *t*-test. **(C-G)** Whole-genome sequencing of the SS14-DC^KO and WT strains (passage #9) post-transformation. **(C)** SS14-DC^KO reads assembled to the WT genome reference (NC_021508.1/CP004011.1) showing a gap where the DC locus previously was. **(D)** SS14-DC^KO reads assembled to the SS14-DC^KO reference genome, where the DC locus was replaced *in silico* with the *kan*^R cassette sequence showing reads aligning to the cassette sequence. **(E)** Schematic of the 51 deleted DCs, color-coded based on their target V region and the *kan*^R cassette (*tp0574* promoter- *kan*^R) that replaced the DCs. **(F)** Reads from the WT SS14 genome aligned to the SS14 reference genome showing the integrity of the DC locus. **(G)** Reads from the WT genome aligned to the SS14-DC^KO reference genome showing a gap where the *kan*^R cassette is located. **(H-I)** Messsage quantification of the *tp0131* and *kan*^R genes in the WT and the SS14-DC^KO strain showing lack of transcriptional polar effects due to the substitution of the DCs. mRNA was reverse-transcribed using either gene-specific primers (GSP, white bars) or random hexamers (RE; black bars); ns: not significant.

*T. pallidum* genome. A search for reads matching the suicide vector backbone yielded no results, confirming lack of residual suicide vector DNA. These results further confirmed elimination of 96% of the DCs (schematically represented in Fig 3E) in the SS14-DC^KO strain and their replacement with the *kan*^R cassette. Furthermore, these data showed that despite this deletion, mutant treponemes are not impaired in their ability to express *tprK* compared to the WT strain. The WT parent strain propagated in parallel to the mutant strain was also sequenced as a control. As expected, in the WT strain the DC locus is intact (Fig 3F), while a gap is visible where the *kan*^R cassette is located when reads from the WT strain were assembled to the KO reference (Fig 3G). A thorough comparison of the WT and KO genomes to evaluate the presence of additional "background" mutations that could have arisen in the KO strain showed no differences between the WT and KO strains outside of the DC locus.

To assess whether the replacement of the DCs induced transcriptional modifications of downstream genes that, in turn, could affect the outcome of the *in vivo* experiments, we quantified the mRNA of the *tp0131* gene in both the WT and mutant strains. In the WT, the *tp0131* gene immediately follows the DCs, and both this gene and the DCs are in the reverse orientation relative to the genome (Fig 3E). In the mutant strain, *tp0131* remains immediately downstream of the inserted transgene, but in reverse orientation, while the *kan*^R cassette is in forward orientation (Fig 3E). All quantifications were normalized to the mRNA of the *tp0574* gene. Quantification of the *kan*^R mRNA was also performed as a control, as the *tp0574* minimal promoter also drives transcription of the resistance cassette in the mutant strain, but no *kan*^R gene is present in the WT. Results (Fig 3H and 3I) showed that expression of *tp0131* was not significantly different in the two strains, regardless of whether the transcript was reverse transcribed using a *tp0131*-specific primer (GSP) or random hexamers (RE). These results support that *tp0131* transcription is not affected by the replacement of the DCs and that there is no transcription of the *tp0131* non-coding strand (possibly originating from the *kan*^R cassette-associated promoter), which reverse transcription with RE would have captured. Overall, however, because *tp0131* transcription is not significantly different in these strains, these data support the absence of polar transcriptional effects in the SS14-DC^KO strain due to the insertion of the resistance cassette. As seen in our previous study [24], the *kan*^R message in the mutant strain was comparable, albeit slightly lower, to that of *tp0574* (Fig 3I).

## Early lesion development is attenuated in rabbits infected with treponemes lacking most DCs

Differences in early lesion development were assessed in New Zealand White rabbits infected intradermally (ID) in multiple sites on their shaved backs with either the WT or mutant strain. Following infection of two groups of four rabbits each, animals were regularly monitored to measure lesion diameter. Samples were collected every ~10 days for quantification of treponemal burden (both biopsies and aspirates) and *tprK* deep sequencing (biopsies only). Serum samples were collected weekly for syphilis serology using both a non-treponemal (Venereal Disease Research Laboratory; VDRL) and treponemal (*Treponema pallidum* particle agglutination; TPPA) test. The diameter of primary lesions in rabbits infected with the WT strain became significantly larger ($p < 0.0001$) compared to the lesions induced by the SS14-DC^KO strain starting at day 9 post-inoculation (Fig 4A). Lesion size in SS14-DC^KO -infected rabbits plateaued at day 50 post-inoculation with an average diameter of 7.0 mm. In control animals, lesion size steadily increased throughout the experiment (Fig 4A) up to day 65 post-inoculation ($p > 0.05$ for lesion diameter values between day 65 and day 72 in WT-infected animals) with mean diameters of 26.75 mm (day 65) and 30.3 mm (day 72), respectively. Pictures of the four animals still in the study at day 72 post-inoculation (Fig 4B) show that in addition to

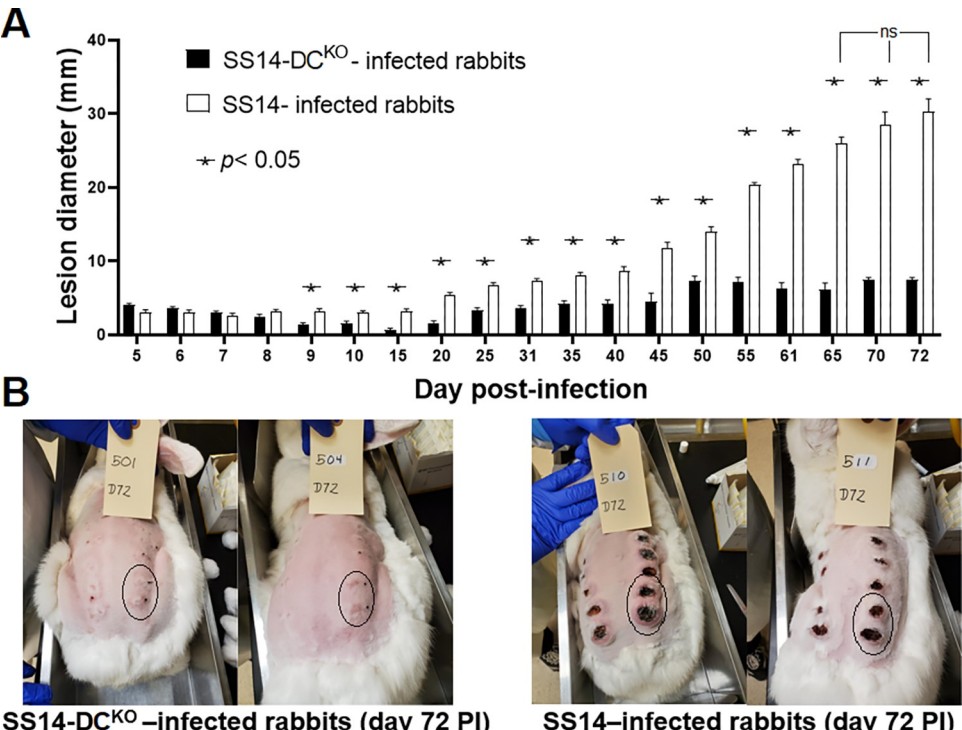

**Fig 4. Lesion progression in infected animals.** (A) Measurements of lesion diameter (mm) over time in rabbits infected with the SS14 strain and the SS14-DC$^{KO}$ strain. Statistical analyses were performed using one-way ANOVA with the Dunnett correction of multiple comparisons or Student's *t*-test, with significance set at $p<0.05$, indicated by an asterisk. The designation "ns" indicates lack of significance in lesion diameter of rabbits infected with the WT strain between day 65 and subsequent time points. (B) Pictures of the animals still in the study at day 72 post-infection showing large, ulcerated lesions in control rabbits (right panel), and significantly smaller ($p<0.05$) lesions in animals infected with the knockout strain (left panel). Circled areas represent lesions that were not yet biopsied at the time the picture was taken. Black ink dots near the lesion area mark the injection sites.

being significantly enlarged, all lesions in the control animals had progressed to an ulcerated state, with large central necrotic areas. In animals infected with the SS14-DC$^{KO}$ strain, ulceration was not evident. Circled areas correspond to lesions that were not biopsied at the time the picture was taken.

## In SS14-DC$^{KO}$-infected rabbits, treponemal burden is reduced and development of humoral immunity to infection is delayed

Treponemal burden in lesion biopsies and needle aspirates was assessed by qPCR and DFM, respectively. In both cases (Fig 5A and 5B), significantly more treponemes were detected in samples from the control animals compared to those from rabbits infected with the SS14-DC$^{KO}$ strain. Although qPCR (Fig 5A) detected *T. pallidum* in all tested samples, treponemal burden in lesions induced by the mutant strain decreased by one (day 9–37), two (day 65–79), or three orders of magnitude (day 37–58) between the control and test samples. As expected, treponemal burden decreased in lesions from control animals over time due to immune clearance [13]. In contrast, the overall burden in lesions from rabbits infected with the SS14-DC$^{KO}$ strain remained stably low (330.6 copies/reaction in average) in samples collected at day 37–79 post-inoculation. No treponemes were detected by DFM of needle aspirates collected from SS14-DC$^{KO}$-infected rabbits at any time point, while treponemes could be detected in lesions from WT-infected animals from day 37–72 post-inoculation (Fig 5B),

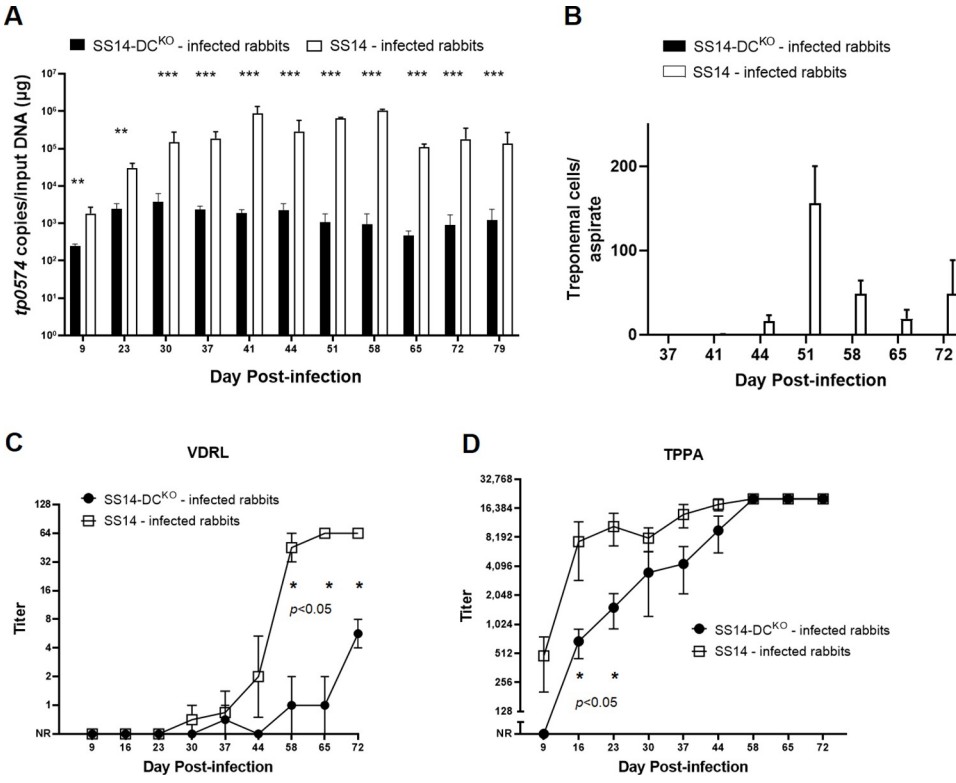

**Fig 5. Treponemal burden and serologies of infected animals. (A)** Treponemal burden in lesion biopsies collected from control and test animals assessed by qPCR targeting the *tp0574* gene. The presence of *T. pallidum* DNA was confirmed in all samples, but treponemal burden in control samples was uniformly higher. Statistical analyses were performed using Student's *t*-test, with significance set at $p < 0.05$ ($p < 0.01$ is indicated with two asterisks, $p < 0.001$ is indicated with three asterisks). **(B)** Treponemal burden in lesion biopsies collected from control and test animals assessed by DFM in lesion needle aspirates. Treponemes could be detected only in samples from control rabbits. **(C)** Significant delay in development of VDRL antibody titers in sera collected from animals infected with the knockout strain compared to serum samples from control rabbits. Controls seroconverted earlier and achieved significantly higher titers compared to test animals. Uninfected rabbit sera (screened at animal arrival at the vivarium) pre-inoculation had no VDRL reactivity. **(D)** Initial delay in development of treponemal antibodies (assessed using the TPPA test) in test animals compared to control rabbits. Titer differences became not significant at day 30 post-inoculation. NR: Nonreactive. In both cases significance ($p < 0.05$) was assessed by performing Student's *t*-test. Uninfected rabbit sera (screened at animal arrival at the vivarium) pre-inoculation had no TPPA reactivity.

suggesting that the treponemal burden in samples from test animals was always below the limit of detection of this assay unless rabbits were infected with the WT strain.

VDRL tests performed on serum samples collected over time showed that the WT-infected animals seroconverted earlier and reached higher titers in a shorter time compared to the SS14-DC$^{KO}$ rabbits (Fig 5C). In rabbits infected with the mutant strain, only one rabbit had seroconverted by day 37 post-inoculation (1:2 titer). This titer, however, could not be reproduced at day 44 post-infection (Fig 5C). VDRL titers between the two rabbit groups remained significantly different from day 58 to the end of the experiment. TPPA titers were significantly different between control and test rabbits only up to day 23 post-infection. Together, these data support impairment of the SS14-DC$^{KO}$ strain's ability to proliferate in dermal lesions, leading to an overall lower treponemal burden and to a delayed development of VDRL and TPPA antibodies.

## Rabbits used as recipients of lymph nodes from SS14-DC^KO-infected rabbits did not develop syphilis infection

Immediately after euthanasia, popliteal lymph nodes were harvested from each of the ID-infected rabbits and transferred via IT injection into a naïve rabbit to assess *T. pallidum* presence. A total of four animals (two per group) were picked randomly and euthanized on day 44 post-infection. The remaining rabbits were euthanized on day 79 post-infection. Rabbits were monitored weekly for orchitis development and bled monthly to assess seroconversion, unless orchitis developed and was confirmed by visualization of treponemes in testicular aspirates, at which point the animal was deemed positive and euthanized. One animal that received lymph nodes from the control rabbits (Table 2) had developed orchitis (confirmed by treponemal presence in a needle aspirate) at day 33 post-inoculation, while two more in the same group developed orchitis (Also DFM-positive) at day 35 post-inoculation. The fourth rabbit in the group developed orchitis at day 74 post-inoculation (Table 2). In contrast, none of the recipients of lymph node extracts from rabbits infected with the SS14-DC^KO strain developed orchitis or seroconverted during the 3-month observation period, supporting absence of treponemal cells in these lymph node extracts at the time the RIT was performed.

## SS14-DC^KO treponemes do not generate new TprK V region variants

Deep sequencing was performed to define TprK isogenicity and antigenic identity in the *T. pallidum* inocula (for both the WT and SS14-DC^KO strains) used for rabbit ID infection and to monitor evolution of diversity over time in progressing early lesions. Table 3 shows the most abundant (>5% of the total for either strain) amino acid sequences for each V region for the WT and SS14-DC^KO strains, while all the detected inoculum variants are reported in S1 Table.

Results showed that neither inoculum was isogenic for *tprK*. Among all V regions, however, V4 exhibited the highest degree of antigenic identity with a single variant representing 96.9% and 96.1% of all detected sequences in the SS14 and the SS14-DC^KO strains, respectively (Tables 3 and S1), while one V1 variant accounted for 96.0% and 81.3% of all sequences found in the WT and mutant strain, respectively. Both strains shared at least one highly represented (>40%) V region sequence in V2, V3, and V7. Higher heterogeneity was found in V5 and V6 (Tables 3 and S1) compared to all the other V sequences.

**Table 2. Rabbit infectivity test results.**

| Infecting strain | RIT day | Donor Rabbit[1] | Time to positive orchitis/DFM[2] | Month 1 serology (day 30 post-RIT) | | Month 2 serology (day 60 post-RIT) | | Month 3 serology (day 90 post-RIT) | |
|---|---|---|---|---|---|---|---|---|---|
| | | | | VDRL[3] | TPPA[4] | VDRL[3] | TPPA[4] | VDRL[3] | TPPA[4] |
| **SS14-WT** | Day 44 post-IDI | 505 | 74 days | NR | NR | NP | NP | | |
| | | 512 | 33 days | NP | NP | | | | |
| | Day 79 post-IDI | 510 | 35 days | R; 1:20 | R; 1:1280 | | | | |
| | | 511 | 35 days | R; 1:4 | R; 1:80 | | | | |
| **SS14-DC^KO** | Day 44 post-IDI | 507 | N/A | NR | NR | NR | NR | NR | NR |
| | | 503 | N/A | NR | NR | NR | NR | NR | NR |
| | Day 79 post-IDI | 501 | N/A | NR | NR | NR | NR | NR | NR |
| | | 504 | N/A | NR | NR | NR | NR | NR | NR |

[1]Identification number of ID-infected donor rabbit. Naïve RIT rabbits received lymph-node from the donor rabbits.

[2]In rabbits with orchitis, treponemes were also detected in testicular needle aspirates using DFM.

[3]VDRL results (NR = Non-reactive; R = Reactive) and titers are shown. NP: Not performed

[4]TPPA results (NR = Non-reactive; R = Reactive) and titers are shown when measured. NP: Not performed.

**Table 3. TprK variability profile in the SS14 and SS14-DC^KO strains used for rabbit ID inoculation.**

| V Region | Amino acid sequence | WT SS14 strain Percent sequence | SS14-DC^KO strain Percent sequence |
|---|---|---|---|
| V1 | GIASETGGAGALKH | 96.06195 | 81.30797 |
| V1 | GIASEKNGGAQPLKH | Not detected | 14.59834 |
| V2 | WEGKDSKGVVQAGANHSK | 85.1821 | 40.83901 |
| V2 | WEGKSNTGAPAAGANHSK | 7.328483 | Not detected |
| V2 | WEGKDSQGKAPAGANHSK | 1.914901 | 25.66107 |
| V2 | WEGKSNTGVVQAGANHSK | 1.042542 | 29.81366 |
| V3 | TLSGDYATARAGADDILWD | 88.95463 | 65.03622 |
| V3 | TLSGGYATARAGADDILWD | 5.402198 | 0.476627 |
| V3 | TLSGDYAQAAGAGADDILWD | Not detected | 28.20839 |
| V4 | TDVGRKKDGAQGTV | 96.91317 | 96.16694 |
| V5 | QASNVFQGVFLTDTTPMLQHDC | 53.176 | 1.716312 |
| V5 | QASNVFQGVFLTTPMQKDDC[1] | 26.09685 | 0.443286 |
| V5 | QASNVFQGVFLTTPMLQHDC[1] | 12.33243 | 68.85708 |
| V5 | KASNVFKDVFLTNAMDMQTHDC | 1.511896 | 12.80529 |
| V5 | KASNVFKDVFLTDTTPMQTHDC[1] | Not detected | 10.32818 |
| V6 | PVHWKALAPAQPPARVDIY | 49.2956 | Not detected |
| V6 | PVHWKALARAQPPARVDIY[1] | 27.46295 | Not detected |
| V6 | PVHWKALARARPPVPAIY[1] | Not detected | 75.82605 |
| V7 | YGGTNKKNDAAPAAPATKWKAEYCGYY | 56.19608 | 41.41053 |
| V7 | YGGTNKKNDAAPATKWKAEYCGYY | 38.03072 | 3.803587 |
| V7 | YGGTNKQAATKWKAEYCGYY | Not detected | 46.71209 |

[1]Synthesized as peptides for ELISA.

TprK profiling in lesion samples collected sequentially showed that generation of new variants was impaired in SS14-DC^KO treponemes compared to WT treponemes for all V regions. Heatmaps showing amino acid sequence variants detected at greater than 0.25% prevalence in inoculum and biopsies are reported in Fig 6 for V6, and in S1–S6 Figs forV1-V5, and V7, respectively. For each V region, a link to an interactive version of the graph is reported in the relative figure legend. In V1, for example (S1 Fig), five non-inoculum variants could be detected in rabbits infected with the WT strain, while no new variants were detected in animals infected with the SS14-DC^KO strain. In V5 and V6 (Figs S5 and 6) a total of 35 and 172 non-inoculum variants, respectively, were detected at greater than 0.25% prevalence in rabbits infected with the WT strain over the course of the experiment, but only 5 and 1 variants were detected in samples from animals that received the mutant strain for the same V regions. All variants detected at all time points are reported in S1 Table. Based on the results shown in Figs 6 and S1–S6, eleven putative new variants were detected in rabbits infected with the SS14-DC^KO strain. These variants, however, were also present in the inoculum, but their prevalence was <0.25% (S3 Table). Taken together, these data support that elimination of the DCs significantly affected the generation of new TprK variants in the SS14-DC^KO. In addition, the two DCs downstream of the *tp0136*, predicted to preferentially recombine into V7 [17], did not contribute to generation of variability in the samples we analyzed from rabbits infected with the SS14-DC^KO.

## Development of humoral reactivity to V region variants is limited in SS14-DC^KO-infected rabbits

Synthetic peptides based on V5 and V6 sequences found in the inoculum treponemes for the WT and the SS14-DC^KO strains (Table 3) were used to perform ELISAs with sera from infected

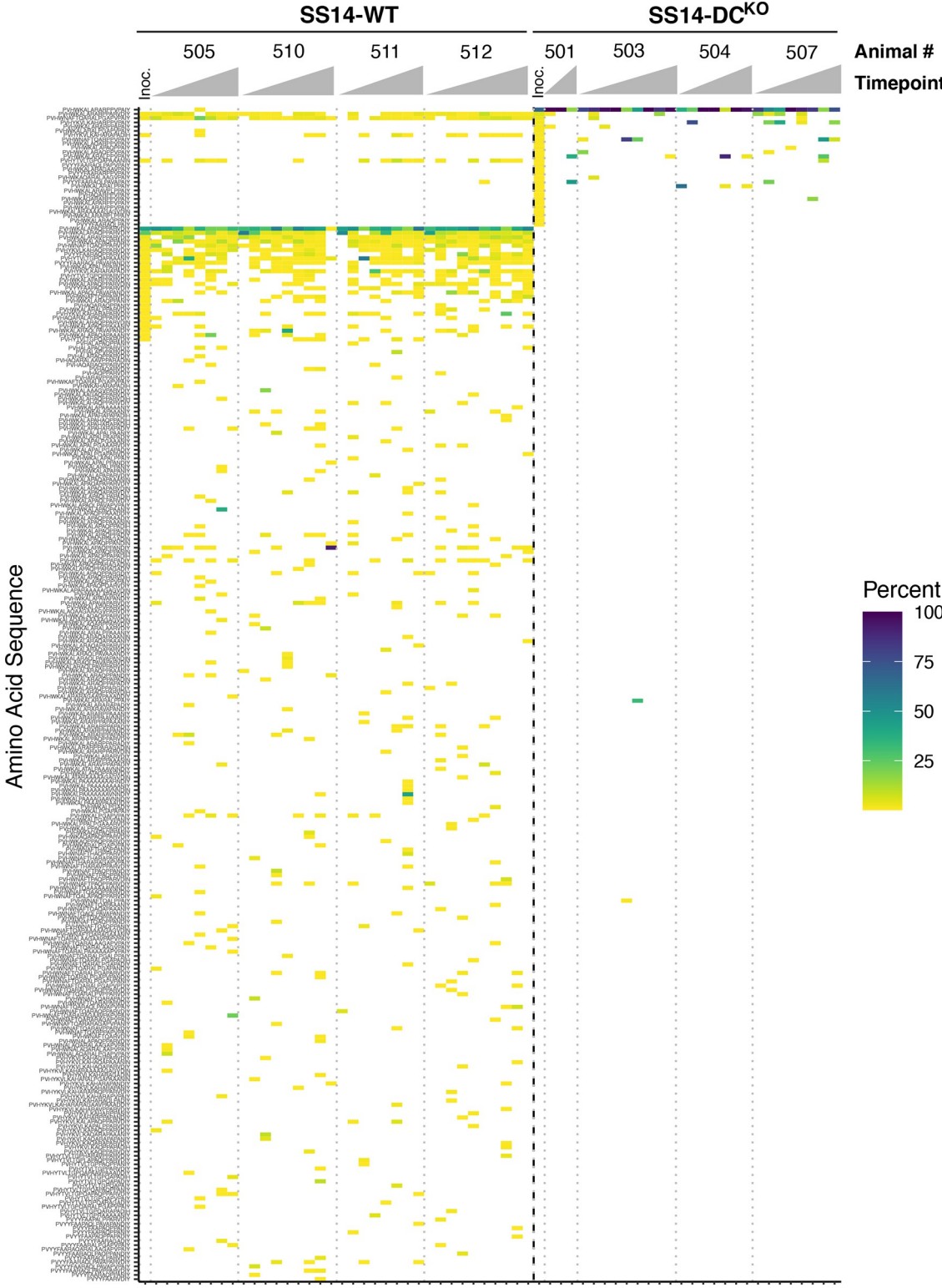

**Fig 6. Heatmap of sequence diversity in TprK V6.** Amino acid sequences of TprK V6 derived from SS14-WT (left) or SS14-DC^KO (right), separated by a bold dotted line. Each column represents the sequences present in a single biopsy for a single animal at a particular timepoint, with biopsy timepoints arranged left to right and animals separated by light dotted lines. Color scale represents the prevalence of each amino acid sequence within each sample. Inoculum samples are labeled "Inoc". Sequences are arranged in order of decreasing prevalence in SS14-DC^KO inoculum, then by SS14-WT inoculum. Missing samples did not yield data. Interactive

heatmaps are available at https://github.com/greninger-lab/Impaired-TprK-Antigenic-Variation. Peptide sequences and prevalence are also reported in S2 Table.

rabbits to evaluate whether variation in V sequence abundance would correlate with the development of a humoral immune response against these sequences. Results showed that in rabbits infected with WT treponemes, selected V regions underwent an initial expansion relatively early following treponemal ID injection compared to the inoculum (Fig 7A–7C). Sequence abundance, however, decreased sharply over the following 2–3 weeks and remained low throughout the remaining timepoints. Antibodies specific for these V regions could be clearly detected (Fig 7D–7F) at day 35 post-infection and, in some instances also at day 28 post-infection, albeit only slightly above background and not for all samples. Reactivity to a recombinant portion of TprK, corresponding to about a third of the protein, and containing two conserved

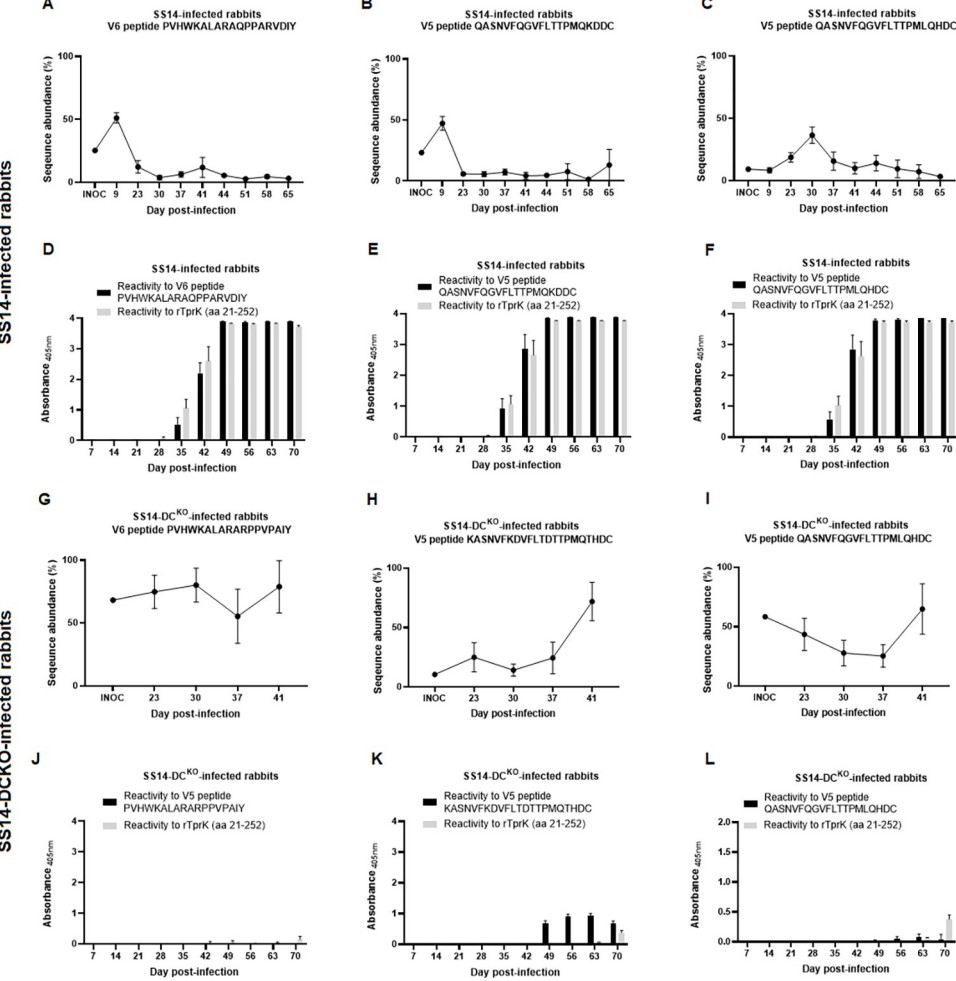

**Fig 7. Comparison between changes over time in sequence abundance of selected V5 and V6 regions present in the inoculum treponemes and the development of humoral immunity to the same regions, assessed by ELISA performed with sera from infected rabbits and synthetic peptides.** Panels A-C (sequence abundance) and D-F (humoral response to the selected peptide) pertain to rabbits infected with the WT strain, while panels G-I (sequence abundance) and J-L (humoral response to the selected peptide) pertain to rabbits infected with the SS14-DC^KO strain. Sequence abundance was based on the *tprK* deep sequencing results for all rabbits.

loops (C1 and C2; Fig 1) was used as a control to evaluate development of immunity to the TprK antigen. These results support that the decrease in V region abundance is due to immune clearance of treponemes carrying such sequences, as also shown in earlier studies [21]. On the contrary, in rabbits infected with the SS14-DC^KO^ strain (Fig 7G–7I) the relative abundance of the analyzed V regions either remained virtually constant or tended to increase over the first 41 days post-inoculation. No deep sequencing data could be uniformly obtained from SS14-DC^KO^ samples past day 41, due to the low treponemal burden in these samples. Antibodies towards these abundant V regions were detectable in sera from the animals infected with the KO strain only at much later time points after infection and at a much lower level compared to the sera from the WT-infected rabbits. Also the reactivity to the larger recombinant TprK fragment became evident only toward the latest time points and never reached a level comparable to that of the control animals (Fig 7J–7L). These results are consistent with a slow host response to the pathogen in these rabbits due to the reduced treponemal burden at the injection site following inoculation (Fig 5A and 5B) which, in turn, induced a host response less robust than by rabbits infected with WT treponemes (Fig 7; as well as lesion size in Fig 4A, and TPPA and VDRL results in Fig 5C and 5D).

## Discussion

The significant impact of syphilis on public and global health warrants research efforts to improve our understanding of the molecular mechanisms behind its pathogenesis. In turn, such efforts could aid strategies for disease control and even vaccine development. The approach to derive *T. pallidum* knockouts recently developed and validated by our groups [24] provides a novel tool to study the role of the pathogen's known or putative virulence factors by comparing loss-of-function mutants with their parent WT isolates. Even though we are still far from having all the genetic tools needed to fully satisfy Koch's molecular postulates [26] for *T. pallidum*, this approach is a significant step forward from to the use of heterologous bacterial systems transformed to express *T. pallidum* genes and perform gain-of-function assays [27–29].

   With the exception of an isolated study [30], all others that focused on TprK pointed at this antigen as the chief player in pathogen immune evasion and survival during natural and experimental infection [14–18,20–23,31–34]. The TprK-encoding *tp0897* gene, therefore, was considered a prime candidate for ablation. However, despite several independent attempts performed with a pUC57-based suicide vector identical to the one used here (but with homology arms specific for the *tprK* gene) and the same transformation/selection approach, a *tprK*^KO^ isolate could not be derived. Although we cannot exclude that these experiments were not successful due to the difficulties of recovering treponemes post-transformation, the hypothesis that *tprK* could be an essential *T. pallidum* gene should not be dismissed. It is intriguing that, even when deep sequencing of full-length *tprK* ORFs is performed [34], amplicons containing non-functional *tprK* sequences (e.g., due to frameshifts or stop codons within the ORF) are not found despite the heavy recombination activity occurring at this site. Aside from immune evasion, therefore, TprK could also mediate a biological function needed for *T. pallidum* survival even *in vitro*, where immune pressure is absent. Preliminary structural [14] and modeling data using AlphaFold2 (Fig 1) support this antigen as a porin transporter, a function predicted for virtually all of the Tpr proteins of *T. pallidum*, even though the nature of the transported substrate(s) is yet to be determined. This study also demonstrated that the putative *tp0127* gene is non-essential in *T. pallidum*. This small 229 codon ORF encodes a hypothetical protein with unknown function, and the gene is reported as the member of a family of paralogs that includes *tp0314, tp0315, tp0346, tp0347, tp0479, tp0617-0619, tp0697, and tp0698*, a group of short, annotated genes encoding hypothetical proteins that could possibly perform the same

function of *tp0127*. This work, therefore, albeit minimally, contributes to defining the essential *T. pallidum* genome.

Despite the difficulty in deriving a *tprK*$^{KO}$ mutant, however, a strain lacking the ~4kbp region containing 96% of the DCs was readily attained, and this deletion did not produce any notable changes SS14-DC$^{KO}$ proliferation *in vitro* or following rabbit IT infection compared to the parent strain. After the weeks needed for the mutant treponemes to recover post-transformation, the *in vitro* growth curve of the SS14-DC$^{KO}$ strain overlapped that of its parent SS14 isolate (Fig 2A), while similar time to orchitis and identical treponemal yields were measured following rabbit infection in testes prior to the point when sufficient cellular infiltrates would induce treponemal clearance. Altogether, these data support that the SS14-DC$^{KO}$ strain is not impaired in its ability to proliferate until a robust host immune response to the pathogen develops. Also in agreement with this conclusion is the evidence that, aside from the targeted DC-containing region, a thorough genome-wide analysis of the WGS data did not reveal any additional genetic difference between the WT and the SS14-DC$^{KO}$ strains, ruling out that growth in absence of the host response, or the *in vivo* attenuated phenotype of the SS14-DC$^{KO}$ strain could be due to "background" mutations. Evidence that the *tprK* mRNA level (Fig 3B) is not affected *in vitro* by DCs ablation in the mutant strain was expected due to the significant distance (~175 kbp) between the *kan*$^{R}$ insertion site and the *tprK* expression site. This evidence further supports that the attenuated dermal lesion phenotype was not due to impaired expression of a hypothetically essential gene. The lack of polar transcriptional effects due to the insertion of the *kan*$^{R}$ cassette was anticipated, as the suicide vector to ablate the DCs was designed to introduce the transgene in the reverse orientation with respect to *tp0131*, the gene downstream of the DCs. *tp0131* (*tprD*) encodes a putative OMP porin, and it possibly has a critical role in *T. pallidum* biology, albeit only partially explored. Therefore, altering the expression of this gene could be detrimental to the pathogen fitness in the host. Although the location of the *tp0131* (*tprD*) promoter is uncertain, this promoter is certainly not within the deleted DC region, as the *tprD* gene in an inverted orientation compared to *kan*$^{R}$. Transcription of the *kan*$^{R}$ gene past the transgene could have hypothetically generated an interfering RNA from the *tp0131* non-coding strand. However, the comparison of transcription levels attained using either RE or GSPs (Fig 3H) did not support this scenario. These data would agree with the presence of a transcription terminator immediately downstream of the *kan*$^{R}$ cassette within the *tp0131* non-coding strand, (with sequence GGAGAAACCAGGGGCATAGCGAGTGTC CTTTTCTGAATAG; stem nucleotides underlined) predicted *in silico* by Erpin and RNAmotif (http://rssf.i2bc.paris-saclay.fr/toolbox/arnold/index.php). This terminator is possibly responsible for ending transcription from the resistance cassette promoter. More comprehensive mRNA sequencing studies using the WT and SS14-DC$^{KO}$ strains propagated both *in vitro* and *in vivo* before immune clearance are currently ongoing to evaluate possible differences in these strains' global transcriptome.

Previous studies also determined that not all *T. pallidum* strains share the same repertoire of DCs and that, even within the Nichols strain, there are lineages missing a portion of the DCs contained in the region downstream of the *tprD* gene [15]. Although the minimum number of DCs necessary for *T. pallidum* to maintain its full virulence is still undetermined, our results showed that that the two remaining DCs downstream of *tp0136* in the SS14-DC$^{KO}$ strain were not sufficient to rescue the attenuated phenotype. In our study, these two sites did not appear to have contributed to the generation of new TprK variants, suggesting that their role in antigenic variation might be marginal or that variation in V7 might have a role during late-stage infection, rather than earlier. Past studies that analyzed the DC molecular architecture to define in which V region a given DC would preferentially recombine into [15,16,34], supported the specificity of the two residual DCs in the SS14-DC$^{KO}$ for V7. Although more

studies are needed to evaluate the importance of these isolated cassettes in generating new TprK variants, it would be plausible to assume that limited generation of variability within one single V region would not fully compensate for lack of variability in all remaining TprK V regions.

A limitation of this study could be seen in the fact that rabbits in both groups were not infected with strains completely isogenic for TprK but rather with a pathogen population carrying very few dominant TprK variants and several low-prevalence variants identified by deep sequencing. This experimental design, however, allowed us to appreciate how underrepresented variants can expand as the more abundant ones are preferentially targeted and cleared during infection by the host response. We are nonetheless deriving a SS14-DC^KO strain fully isogenic for TprK to evaluate the hypothesis that, upon ID infection with this strain, early disease manifestations will be even more accentuated compared to the present study. The derivation of a clonal strain will also help shed light on the role of the two residual DCs for *tprK* present in the SS14-DC^KO genome. Although these two DCs do not seem to contribute to diversity generation according to the results of this initial study, the possibility of removing the genomic region harboring these residual sequences will need to be considered for future experiments if evidence that recombination in V7 occurs in the mutant strain.

The sensitivity of the RIT assay, albeit recently questioned [35], is known to be very high, with a 50% infectious dose of 23 spirochetes [36,37]. The lack of seroconversion of RIT rabbits injected with lymph node extracts obtained at both day 44 and 79 from rabbits infected ID with the SS14-DC^KO strain is intriguing, particularly if we consider that full treponemal clearance was not supported by qPCR on samples collected at both time points at which the RIT was performed. However, one could hypothesize that the significant reduction of the treponemal burden seen in dermal lesions would necessarily translate in an abatement of the number of treponemes that had disseminated to the lymph nodes to a number below the sensitivity of the RIT assay. It is also possible that qPCR was detecting genomic DNA from dead treponemes. This hypothesis would be supported not only by the RIT results, but also by the evidence that *tprK* profiling, which requires amplification of the *tprK* coding sequence, was generally not attainable in samples obtained after day 40 from rabbits infected with the SS14-DC^KO. In the future, we will also collect biopsy specimens for RNA extraction and quantification as a surrogate of treponemal viability and to assess the presence of viable treponemes by inoculating Sf1Ep cultures *in vitro*. In this context, an isogenic strain would also help us understand more clearly whether the host response cleared the SS14-DC^KO treponemes during infection upon performing the RIT assay. Regardless, evidence that the test animals remained RIT negative, when all the control animals developed orchitis or seroconverted, further supports the attenuation phenotype of the SS14-DC^KO strain.

A model for the attenuated phenotype exhibited by the mutant strain could be hypothesized on the evidence that following rabbit ID infection, inoculum SS14-DC^KO treponemes started to be cleared early after inoculation. By day 9, lesion development and treponemal burden in the skin tissue biopsies started to diverge significantly between the test and control rabbits (Figs 4A and 5A). Within a few weeks, treponemal antibodies (including to TprK) had started to develop (Figs 5D and 7D–7F) and possibly contributed to clearance of WT treponemes expressing the most abundant TprK inoculum sequences (Fig 7A–7C). Nonetheless, WT treponemes could still expand (Fig 5A) through the generation of new variants or the expansion of low-frequency ones in the inoculum (Figs 6 and S1–S6). In rabbits infected with the SS14-DC^KO strain, however, treponemal burden increase was likely hindered by the lack of DCs, even though underrepresented variants could still expand (Figs 6, and S1–S6 and S2 Table) and maintain a very low pathogen burden over time (Fig 5A). Abundant *tprK* inoculum sequences in the SS14-DC^KO treponemes were virtually absent by day 41 post-inoculation

(Fig 7G–7I), even though the delayed immunity that developed to the infection in these animals likely made the antibodies responsible for clearance detectable only at later time points post-infection (Fig 7J–7L). Future experiments will evaluate whether immunosuppression of the animals prior to infection would rescue the attenuation phenotype exhibited by the SS14-DC$^{KO}$ strain.

## Conclusions

We demonstrated that when genetically altered to impair its ability to vary the TprK antigen, the *T. pallidum* SS14 strain induces attenuated disease manifestations and is unable to proliferate in the rabbit model of ID infection as effectively as the WT strain. This work further supports the importance of TprK for *T. pallidum* virulence and emphasizes how our newly developed genetic engineering approach can contribute to shed light on the molecular mechanisms behind syphilis pathogenesis.

## Materials and methods

### Ethics statement

Male NZW rabbits (*Oryctolagus cuniculus*) ranging from 3.5–4.5 kg in weight were purchased from Western Oregon Rabbit Company (Philomath, OR) and housed at the Harborview Medical Center–Research and Training Building vivarium. Care was provided in accordance with the Guide for the Care and Use of Laboratory Animals [38]. All procedures involving these animals were approved by the UW Institutional Animal Care and Use Committee (IACUC; Protocol # 4243–01, PI: Lorenzo Giacani). Upon arrival, rabbits were allowed to acclimate to the vivarium for several days, then bled and tested with both a treponemal (*Treponema pallidum* particle agglutination, TPPA; Fujirebio, Tokyo, Japan) and a non-treponemal (Venereal Disease Research Laboratory, VDRL; Becton Dickinson, Franklin Lakes, NJ) test to confirm lack of previous or current infection with the agent of rabbit syphilis, *Treponema paraluiscuniculi*. Both tests were performed according to the manufacturer's instructions. Only seronegative rabbits were used for experimental infection with the SS14-DC$^{KO}$ and SS14 parent strains (see paragraph below for rabbit infection). Rabbit rooms in the vivarium are permanently kept at 65˚F. Animals were only given antibiotic-free food and water.

### TprK structure modeling

We used the ColabFold interface [39] to construct Multiple Sequence Alignments (MSA) for the TprK query sequences by searching UniRef30 [40], Mgnify [41] and ColabFold sequence databases with MMSeq2 [42]. The MSA was used as input for structure prediction with AlphaFold2 [43] using the default settings (template = False, amber relax = False, 3 recycles). Increasing the number of recycles to 6 did not significantly improve the prediction. Visualization was performed using PyMol software (https://pymol.org*) [44].*

### Suicide vector construction

The pUC57 vector (Genscript, Piscataway, NJ) was engineered to carry the *kan*$^R$ gene downstream of the promoter and ribosomal binding site (RBS) of the *T. pallidum tp0574* gene (encoding the Tp47 lipoprotein) as described by Weigel *et al.* [45]. An 8-nt spacer separated the RBS and the start codon of the *kan*$^R$ gene. The *kan*$^R$ gene sequence was derived from the Rts1 plasmid originally isolated from *Proteus vulgaris* [46] and chosen because the CDC STI treatment guidelines do not recommend the use of kanamycin for syphilis therapy. Codon optimization of the *kan*$^R$ gene was shown not to be necessary in a previous study [24].

Two homology arms corresponding to the regions flanking the 51 DCs located downstream of the *tprD* gene were cloned to the left and right of the *tp0574* promoter-*kan*[R] hybrid sequence in the pUC57 vector, respectively. The upstream and downstream arms were 990 bp and 965 bp in length, respectively, and corresponded to position 149,425–148,434 and 152,401–153,366 of the SS14 genome. The construct was cloned between the EcoRV and KpnI sites of the pUC57 vector, in reverse orientation compared to the *lac* promoter upstream of the poly-linker. Prior to use, the insert underwent Sanger sequencing to ensure sequence accuracy. The sequence of the insert is provided in S1 File. This construct was named p*DC*arms-47p-*kan*[R]. Primers annealing a) to the vector only, b) within the cloned insert, and c) upstream of the DCs homology arms in the *T. pallidum* genome are reported in Table 1. Previous work confirmed that sequences flanking the *bla* gene (encoding a β-lactamase) of the pUC57 did not carry sufficient homology to *T. pallidum* DNA to induce recombination and integration of the *bla* gene into *T. pallidum* genome [24].

To obtain a highly concentrated, endotoxin-free plasmid preparation, the p*DC*arms-47p-*kan*[R] was transformed into TOP10 *E. coli* cells (Thermo Fisher, Waltham, MA), which were then grown first in a 5-ml starter culture overnight, and then in 500 ml of LB media supplemented with 100 μg/ml of ampicillin at 37˚C. The plasmid was purified using the Endo-Free Plasmid Mega Kit (Qiagen, Germantown, MD) according to the manufacturer's instructions. Following purification, plasmid concentration was assessed using an ND-1000 spectrophotometer (Nanodrop Technologies, Wilmington, NC). The purified vector was then aliquoted and stored at -20˚C until use.

## Source of *T. pallidum* for *in vitro* cultivation, transformation, and selection

The SS14 parent strain of *T. pallidum* used for *in vitro* propagation was obtained from a frozen stock previously propagated via intratesticular (IT) infection in NZW rabbits as previously reported [47]. The SS14 strain was selected as a representative of the members of a globally dominant cluster, known as the SS14 clade [48]. *In vitro* strain co-culturing with rabbit Sf1Ep cells was performed as per published protocols in a 6-well culture plate (Corning Inc, Corning, NY) [49]. Before the addition of the treponemal cells, Sf1Ep cells were incubated in a 5% $CO_2$ atmosphere in a HeraCell 150 incubator (Thermo Fisher). Following treponemal cell inoculation, the microaerophilic atmosphere (MA; 1.5% $O_2$, 3.5% $CO_2$, and 95% $N_2$) necessary to maintain treponemal viability was generated in a Heracell VIOS 160i tri-gas incubator (Thermo Fisher).

For transformation, treponemes were first sub-cultured into the wells of a 24-well plate as previously reported [24]. Briefly, the day before treponemal inoculation, a 24-well plate was seeded with $2x10^4$ rabbit Sf1Ep cells/well in 2.5 ml of culture media. The plates were then incubated overnight in the HeraCell incubator. On the same day, TpCM-2 media was prepared according to protocol and equilibrated overnight at 34˚C in the tri-gas incubator. The following day, cell culture media was removed from the 24-well plate, and cells were rinsed with TpCM-2 media equilibrated in the tri-gas incubator. Subsequently, each well was filled with 2.5 ml of equilibrated TpCM-2 media, and the plate was transferred to the tri-gas incubator. To prepare the treponemal inoculum, the Sf1Ep cells seeded the previous week with the parent SS14 cells were trypsinized to allow the release and counting of spirochetes using an Optiphot 2 dark-field microscope (DFM; Nikon, Melville, NY). Treponemal counting was performed according to Lukehart and Marra [50]. A total of ~$2.5x10^8$ treponemes were inoculated 24 hours after plating the Sf1Ep. Following treponemal addition, the total volume of media in each well was brought to 2.5 ml. Two days following treponemal cell addition, 1 ml of old medium was replaced with fresh medium. Four days after treponemal cell inoculation, the

culture media was removed gently so as not to disturb adherent treponemes. Media was replaced by 500 µl of transformation buffer (50 mM $CaCl_2$, 10 mM Tris pH 7.4; equilibrated in MA) containing 15 µg total of p*DC*arms-47p-*kan*[R] vector. Cells were incubated in transformation buffer for 10 min at 34˚C in the tri-gas incubator and then washed twice with equilibrated TpCM-2 media to remove free plasmid from the culture wells. Finally, 2.5 ml of fresh TpCM-2 equilibrated in tri-gas were added to the wells, and plates were returned to the tri-gas incubator. The following day, concentrated tissue-culture grade liquid kanamycin sulfate (Sigma-Aldrich, St. Louis, MO) was added to the appropriate wells to reach a final concentration of 200 µg/ml. As a control, to confirm the treponemicidal activity of kanamycin, WT treponemes were also incubated in fresh TpCM-2 media containing kanamycin. Previous work [24] demonstrated lack of toxicity associated to $CaCl_2$ exposure for treponemes, and a $CaCl_2$ only control was not performed here again. The parent SS14 strain, however, was propagated in parallel to the transformed treponemes as a control. Kanamycin-containing TpCM-2 media was exchanged weekly but treponemes were sub-cultured every two weeks as already described [24] until they reached a density of ~$10^7$ cells/ml determined by DFM. At this point, cultures were sub-cultured first into one well of a 6-well plate (seeded with $10^5$ Sf1Ep cells on the previous day) and then into all the wells of a 6-well plate at the following passage to further expand the strain and minimize the chances of culture loss due to contamination. At selected passages, residual treponemes were pelleted by centrifugation at 15,000 rpm for 10 min on a tabletop centrifuge and resuspended in 1X DNA lysis buffer (10 mM Tris-HCl, 0.1 M EDTA, 0.5% SDS) for DNA extraction, or Trizol (Thermo Fisher) for RNA extraction. A WT SS14 strain with limited TprK variability was attained through *in vitro* cultivation of treponemes following limiting dilution [49].

## DNA and RNA extraction

DNA extraction from cultured SS14-DC[KO] or WT strains propagated *in vitro* following the transformation procedure as well as from lesion biopsies (see below paragraph about rabbit infection and sample collection) was performed using the QIAamp DNA Mini Kit (Qiagen) or the Zymo Quick-DNA 96 Kit (Zymo Research, Irvine, CA), respectively, according to the manufacturer's instructions. Extracted DNA was stored at -80˚C until use for qualitative/quantitative amplifications or genome sequencing (see below paragraphs) to evaluate successful insertion of the *kan*[R] gene or to evaluate treponemal burden. Samples for RNA extraction were processed according to the Trizol reagent manual. Total RNA was treated with DNaseI according to the protocol provided with the TURBO DNA-free kit (Thermo Fisher). DNA-free RNA was checked for residual DNA contamination by qualitative amplification using primers specific for the *tp0574* gene (primers in Table 1) as already described [51]. Reverse transcription (RT) of total RNA was performed using the High-Capacity cDNA Reverse Transcription kit (Thermo Fisher) with random hexamers or gene-specific primers (Table 1) according to the provided protocol. cDNA samples were stored at -80˚C until use for qPCR to quantify the level of expression of the *tprK* gene as previously reported [52], and the *tp0131* (*tprD*) gene, also as reported [53]. All transcripts were normalized to the *tp0574* expression level (primers in Table 1).

## Droplet digital PCR (ddPCR)

Droplet digital PCR assays were conducted to assess the ratio between the number of copies of the *kan*[R] gene, the *tp0001* gene (*dnaA*), and an open reading frame (ORF; *tp0127*) encoding a 229 amino acid-long hypothetical protein with unknown function contained within the ablated DCs region. This short ORF was selected because it is unique to the ablated region,

while primers annealing directly to the DCs sequences could potentially also recognize sequences in the *tprK* expression site. Analyzed DNA was extracted from the SS14-DC$^{KO}$ from *in vitro* passages 2–5 post-transformation and from the parent SS14 strain propagated in parallel. ddPCR was performed on a Bio-Rad QX100 system (Bio-Rad, Carlsbad, CA). Primers and probe are listed in Table 1. Each reaction was performed using ddPCR Supermix for Probes (Bio-Rad) with the final concentration of primers at 900 nM and probes at 250 nM in a total reaction volume of 25 µl. Before amplification, template DNA was digested with 25 units of EcoRI (New England Biolabs, Ipswich, MA). After droplet generation, droplets were transferred to a 96-well PCR plate and amplified on a 2720 Thermal Cycler (Thermo Fisher) with the following cycling parameters: 94˚C for 10 min, followed by 40 cycles of 94˚C for 30 s and 60˚C for 1 min, and 98˚C hold for 10 min. After amplification, the plate was transferred to QX200 droplet reader (Bio-Rad). Results were analyzed using the QuantaSoft software (Bio-Rad).

### *T. pallidum* whole-genome sequencing

Whole-genome sequencing was performed following DNA extraction of SS14-DC$^{KO}$ cultured *in vitro* and harvested at passage #9 post-transformation. Pre-capture libraries were prepared from up to 100 ng input DNA using the KAPA Hyperplus kit (Roche) and TruSeq adapters and barcoded primers (Illumina), following the manufacturer's protocols, yielding an average library size longer than 400 bp. Hybrid capture of *T. pallidum* genomic DNA was performed overnight (>16 hours) using a custom IDT xGen panel designed against the reference genome NC_010741, following the manufacturer's protocol. Short-read sequencing was performed on an Illumina NextSeq 2000 with 150 bp paired-end reads, yielding 2.75 and 8.20 million raw reads for WT and mutant strains, respectively. Reads were adapted and quality trimmed using Trimmomatic v0.39 [54] and assembled to the *T. pallidum* SS14 reference genome NC_021508.1, or NC_021508.1 with the DC region manually replaced by the *kan*$^R$ cassette, using Bowtie2 v2.4.1 [55] and deduplicated with Picard MarkDuplicates v2.23.3 (available at http://broadinstitute.github.io/picard), to an average coverage exceeding 177x for the WT and 405x for the KO strain. Manual confirmation of expected coverage and junctions was performed by visual inspection in Geneious Prime v2020.1.2 [56]. Raw reads were deposited to the NCBI Sequence Read Archive under BioProject PRJNA909291.

### Qualitative and quantitative PCR

Samples harvested during routine propagation of the SS14-DC$^{KO}$ and parent strains were assessed for integration of the *kan*$^R$ gene into the DC locus by using qualitative PCR. All primers used in these assays and amplicon sizes are reported in Table 1. In the first amplification, the sense primer targeted a region of the *T. pallidum* genome immediately upstream of the left homology arm of the vector (and hence not cloned into p*DC*arms-47p-*kan*$^R$), while the antisense primers targeted the *kan*$^R$ gene, with the rationale that only a *kan*$^R$ gene integrated into the DC region would provide amplification. In the second PCR, the sense primer targeted the *kan*$^R$ gene, and the antisense primer targeted the genomic region downstream of the right homology arm of the vector. A third PCR used both primers external to the homology arms to show a differential in size when DNA from the transformed and parent strain, respectively, was used. Amplification of the *tp0574* gene was used as positive amplification control, and amplification with pUC57 plasmid-specific primers was performed to confirm that no residual transformation plasmid persisted in the SS14-DC$^{KO}$ culture. Amplifications were performed using five microliters of extracted DNA in 50 µl final volume containing 2.5 units of GoTaq polymerase (Promega, Madison, WI), 200 µM of each dNTP, 1.5 mM of MgCl$_2$, and 400 nM

of sense and antisense primers. Cycling parameters were initial denaturation (94°C) and final extension (72°C) for 10 min each. Denaturation (94°C) and annealing (60°C) steps were carried on for 1 min each, while the extension step (72°C) was carried out for 1 or 2 min depending on amplicon length. A total of 40 cycles were performed in each reaction.

A TaqMan qPCR assay targeting the *tp0574* gene was instead used to quantify treponemal load in samples extracted from rabbit lesion biopsies as previously described [57]. Primers/probe and amplicon size are reported in Table 1. Amplifications were run on a QuantStudio 3 or 5 thermal cycler (Thermo Fisher) and results were analyzed using the instrument software. Data were imported into Prism 8 (GraphPad Software, San Diego, CA) and further analyzed to assess statistical significance of the values from the control rabbits and those infected with the knockout strain using one-way ANOVA with the Dunnett test for correction of multiple comparisons or Student's *t*-test, with significance set at $p < 0.05$ in both cases.

### TprK molecular profiling

TprK libraries were prepared similarly as previously described [17]. Technical duplicates were performed with thirty-five cycles of PCR in a 50 μL reaction with 0.4 mM TprK forward and reverse primers (S4 Table) using the 2x CloneAmp MasterMix (Takara) with a 62°C annealing temperature. Biopsy DNA input was limited to 4 μL per reaction due to high host DNA concentration inhibiting amplification, which represented a range of 75–1433320 genome copies. For the parent and SS14-DC$^{KO}$ inoculum propagated *in vitro*, 1e6 genome copies were input. PCR products were run on a 1% agarose gel to confirm the presence of a band at approximately 1.6 kb and cleaned with 0.6x Ampure beads (Beckman Coulter); samples with fewer than 200 input genome copies were excluded from further analysis due to inconsistent amplification. Two nanograms of PCR product were used in a two-fifths Nextera (Illumina) library prep with 15 cycles of amplification. Low molecular weight products were removed with 0.6x Ampure beads, with an average size of approximately 350 bp. Libraries were quantified, pooled, and run on a 2x150 paired end run on an Illumina Nextseq 2000. Raw reads were adapter-trimmed and filtered for quality scores above Q20 using Trimmomatic v0.39 [54], followed by merging the paired end reads using BBmerge (https://www.osti.gov/biblio/1241166) in default parameters. Variable site sequences were identified using custom python and R scripts [22], and those containing frameshifts or nonsense mutations in the translated sequences removed. Replicate samples yielding a minimum of 2,000 reads in all V regions in both replicates were kept for downstream analysis, corresponding to input merged reads ranging between 89182–5746195. Only sequences with 5 or more raw reads per technical replicate were retained per sample. Samples with a Pearson correlation coefficient lower than 0.95 between replicates were removed, and the number of reads for each variable region in each technical replicate were averaged. Sequences with less than 0.25% prevalence per V region per sample were excluded from further analysis. Raw reads are available in NCBI BioProject PRJNA909291.

### Rabbit intradermal infection, lesion monitoring, and sample collection

Two groups of 4 animals each were infected intradermally (ID) on the same day on ten sites/animal on their shaved backs with $10^6$ viable *T. pallidum* cells. One group received the WT SS14 strain, and the other group was infected with the SS14-DC$^{KO}$ strain. Briefly, the day of the ID infection, treponemes were harvested from culture plates as described above for routine passages. Following a low speed-centrifugation for 10 min at 1,000 rpm in a R4702 centrifuge (Eppendorf, Hauppauge, NY) to remove rabbit Sf1Ep cells, treponemal cells were enumerated and checked for motility by DFM and diluted using sterile saline to a final concentration of

$10^7$ cells/ml. One hundred microliters of treponemal suspension were injected in each site on the rabbit backs. Treponemal cells were evaluated again for motility using DFM after performing ID inoculation to ensure that pathogen viability was unaltered at the end of the procedure.

Intradermally-infected rabbits were shaved daily post-inoculation to allow monitoring of lesion progression and to facilitate collection of needle aspirates and lesion biopsies. Two animals from each group were euthanized on day 44 post-inoculation to harvest popliteal lymph nodes and perform an RIT (see paragraph below). The remaining four animals were monitored for an additional 36 days, and then also euthanized to perform RIT.

Lesion biopsies and aspirates were collected at day 9, 22, 30, 37, 41 post-infection from all animals, at day 44 post-infection from the four animals that were euthanized first, and at day 51, 58, 65, 72, and 79 from the remaining four animals. One biopsy per animal was obtained per collection day using a 4 mm biopsy punch. At day 44, however, biopsies were collected from all lesions (unless previously biopsied) of the animals that were euthanized to perform the first round of RIT assays. Aspirates were collected from approximately half of the lesions from each rabbit per collection day. Once biopsied, a lesion was no longer measured or used to collect aspirates. Aspirates were obtained using an Excel 1cc zero dead space tuberculin syringe with a 25Gx3/8" needle (Excel international, Birmingham, AL) by inserting the needle laterally into the side of the developing lesion. Once inserted, the plunger was drawn, and the needle moved back and forth five times while holding the plunger. Lesion aspirates were resuspended in 60 μl of sterile saline, and 18 μl of each suspension were examined by DFM to assess for the presence and number of treponemes. *T. pallidum* cells were counted blindly in a total of 50 fields/sample. The operator was blinded to the identity of the samples. Biopsies were minced using forceps and a sterile scalpel, resuspended in 1X DNA lysis buffer. Serum was obtained weekly from each animal to perform VDRL and TPPA tests to monitor development of humoral immunity in response to infection. Statistical analyses were performed using one-way ANOVA with the Dunnett correction of multiple comparisons or Student's *t*-test, with significance set at $p<0.05$.

## Rabbit infectivity test (RIT)

Immediately after euthanasia, popliteal lymph nodes were harvested from rabbits to perform an RIT as previously described [50]. Following injection of minced popliteal lymph nodes in the animal's left testis, RIT rabbits were monitored twice a week for orchitis development. If orchitis developed, to ascertain that it was due to *T. pallidum* infection, testicular needle aspiration was performed as described above to be analyzed by DFM. If orchitis was confirmed by finding treponemes in the sample, then the animal was euthanized. As orchitis does not always develop in RIT animals, rabbits were also bled monthly to assess seroconversion using VDRL and TPPA tests. If seroconversion occurred, the animal was euthanized. If at the end of the 3-month observation period following IT injection, the rabbit had not developed orchitis or seroconverted, then the RIT outcome was negative, and the animal was deemed uninfected.

## Rabbit intratesticular infection

To assess differences in *in vivo* growth between the parent and knockout strains studied here, one rabbit was infected IT with the *T. pallidum* parent SS14 strain ($7\times10^7$ cells/testis), while a second animal was inoculated with the SS14-DC$^{KO}$ strains IT ($6\times10^7$ cells/testis). Both animals were monitored daily to determine the time to orchitis. At peak orchitis, animals were euthanized, testes were harvested, and content extracted in 10 ml of sterile saline to quantify treponemal burden by DFM. Procedures were performed as previously described [50].

## ELISA using V region synthetic peptides

Synthetic peptides were designed based upon the TprK profiling results for a subset of animals infected with the SS14-DC$^{KO}$. A total of 5 peptides matching inoculum sequences for both the WT and SS14-DC$^{KO}$ strains (Table 3, labelled with a superscripted "1") were produced by Genscript (Piscataway, NJ). Upon reception, lyophilized peptides were rehydrated in sterile PBS (supplemented with DMSO or acetic-acid) to a stock solution of 4mg/ml. Reconstituted peptides were stored at -80°C until use. For ELISA, peptides were further diluted to 10 μg/ml in PBS, and 50 μl of working dilution (500 ng total) were used to coat the wells of a 96-well Microwell Maxisorp flat-bottom plate (Thermo-Fisher, Waltham, MA) as previously described [19]. Briefly, plates were then washed three times with PBS+0.05% Tween-20 and blocked with 200 μl of 5% nonfat dry milk (NFM) in sterile PBS for three hours. At this step, NFM was used to block no-antigen wells for background signal. Wells were then washed again as above and primary antibody was applied immediately thereafter. Sera from infected animals were used at a 1:20 dilution. After overnight incubation with the primary antibody, plates were washed again as above and 100 μl of a goat anti-rabbit IgG (H + L) alkaline phosphatase conjugate (Sigma, St. Louis, MO) diluted 1:2,000 in PBS/1% NFM were added to each well and incubated for 1.5h at room temperature. Plates were washed again and developed with 50 μl/well of 1 mg/ml para-Nitrophenyl Phosphate (pNPP) substrate (Sigma), dissolved in glycine buffer (0.1M glycine, pH 10.4, 1mM $MgCl_2$ and 1 mM $ZnCl_2$) for 1 h according to the manufacturer's instructions. Absorbance was measured at $OD_{405}$ using a Molecular Devices SpectraMax Plus microplate reader (Molecular Devices, San Jose, CA). The mean of triplicate experimental wells minus the mean of pre-infection serum wells tested with the same antigen was calculated for all proteins/peptides tested. Plotted data represent the mean ± SEM for triplicate wells tested. Results were analyzed using Student's *t*-test with significance set at $p \leq 0.05$.

## Supporting information

**S1 Fig. Heatmap of sequence diversity in TprK V1.** Deep sequencing of TprK V1 region showing persistence of inoculum variants and generation of non-inoculum variants in samples collected overtime from rabbits infected with the WT SS14 strain ("WT"-labeled samples, left side of the map) and form rabbits infected with the SS14-DC$^{KO}$ strain ("KO"-labeled samples, right side of the map), separated by a bold vertical dashed line. In each map, inoculum sequences for the WT strain are labeled as "WT Inoc B00 Day 00", where WT Inoc indicates WT inoculum treponemes, B00 indicates that the sample was not obtained from a biopsy, and Day 00 indicates the experiment's time 0. The same nomenclature was adopted for the SS14-DC$^{KO}$ inoculum, with the exception that KO replaced WT. Samples collected post-inoculation report rabbit number (R#), biopsy number (B#), and day post-inoculation the sample was obtained (Day#). Light gray dashed lines separate individual animals. Missing samples did not yield data. The same heatmap in interactive format is available at https://github.com/greninger-lab/Impaired-TprK-Antigenic-Variation. Peptide sequences and prevalence are also reported in S2 Table.
(PDF)

**S2 Fig. Heatmap of sequence diversity in TprK V2.** Deep sequencing of TprK V2 region showing persistence of inoculum variants and generation of non-inoculum variants in samples collected overtime from rabbits infected with the WT SS14 strain ("WT"-labeled samples, left side of the map) and form rabbits infected with the SS14-DC$^{KO}$ strain ("KO"-labeled samples, right side of the map), separated by a bold vertical dashed line. In each map, inoculum sequences for the WT strain are labeled as "WT Inoc B00 Day 00", where WT Inoc indicates

WT inoculum treponemes, B00 indicates that the sample was not obtained from a biopsy, and Day 00 indicates the experiment's time 0. The same nomenclature was adopted for the SS14-DC$^{KO}$ inoculum, with the exception that KO replaced WT. Samples collected post-inoculation report rabbit number (R#), biopsy number (B#), and day post-inoculation the sample was obtained (Day#). Light gray dashed lines separate individual animals. Missing samples did not yield data. The same heatmap in interactive format is available at https://github.com/greninger-lab/Impaired-TprK-Antigenic-Variation. Peptide sequences and prevalence are also reported in S2 Table.
(PDF)

**S3 Fig. Heatmap of sequence diversity in TprK V3.** Deep sequencing of TprK V3 region showing persistence of inoculum variants and generation of non-inoculum variants in samples collected overtime from rabbits infected with the WT SS14 strain ("WT"-labeled samples, left side of the map) and form rabbits infected with the SS14-DC$^{KO}$ strain ("KO"-labeled samples, right side of the map), separated by a bold vertical dashed line. In each map, inoculum sequences for the WT strain are labeled as "WT Inoc B00 Day 00", where WT Inoc indicates WT inoculum treponemes, B00 indicates that the sample was not obtained from a biopsy, and Day 00 indicates the experiment's time 0. The same nomenclature was adopted for the SS14-DC$^{KO}$ inoculum, with the exception that KO replaced WT. Samples collected post-inoculation report rabbit number (R#), biopsy number (B#), and day post-inoculation the sample was obtained (Day#). Light gray dashed lines separate individual animals. Missing samples did not yield data. The same heatmap in interactive format is available at https://github.com/greninger-lab/Impaired-TprK-Antigenic-Variation. Peptide sequences and prevalence are also reported in S2 Table.
(PDF)

**S4 Fig. Heatmap of sequence diversity in TprK V4.** Deep sequencing of TprK V4 region showing persistence of inoculum variants and generation of non-inoculum variants in samples collected overtime from rabbits infected with the WT SS14 strain ("WT"-labeled samples, left side of the map) and form rabbits infected with the SS14-DC$^{KO}$ strain ("KO"-labeled samples, right side of the map), separated by a bold vertical dashed line. In each map, inoculum sequences for the WT strain are labeled as "WT Inoc B00 Day 00", where WT Inoc indicates WT inoculum treponemes, B00 indicates that the sample was not obtained from a biopsy, and Day 00 indicates the experiment's time 0. The same nomenclature was adopted for the SS14-DC$^{KO}$ inoculum, with the exception that KO replaced WT. Samples collected post-inoculation report rabbit number (R#), biopsy number (B#), and day post-inoculation the sample was obtained (Day#). Light gray dashed lines separate individual animals. Missing samples did not yield data. The same heatmap in interactive format is available at https://github.com/greninger-lab/Impaired-TprK-Antigenic-Variation. Peptide sequences and prevalence are also reported in S2 Table.
(PDF)

**S5 Fig. Heatmap of sequence diversity in TprK V5.** Deep sequencing of TprK V5 region showing persistence of inoculum variants and generation of non-inoculum variants in samples collected overtime from rabbits infected with the WT SS14 strain ("WT"-labeled samples, left side of the map) and form rabbits infected with the SS14-DC$^{KO}$ strain ("KO"-labeled samples, right side of the map), separated by a bold vertical dashed line. In each map, inoculum sequences for the WT strain are labeled as "WT Inoc B00 Day 00", where WT Inoc indicates WT inoculum treponemes, B00 indicates that the sample was not obtained from a biopsy, and Day 00 indicates the experiment's time 0. The same nomenclature was adopted for the

SS14-DC$^{KO}$ inoculum, with the exception that KO replaced WT. Samples collected post-inoculation report rabbit number (R#), biopsy number (B#), and day post-inoculation the sample was obtained (Day#). Light gray dashed lines separate individual animals. Missing samples did not yield data. The same heatmap in interactive format is available at https://github.com/greninger-lab/Impaired-TprK-Antigenic-Variation. Peptide sequences and prevalence are also reported in S2 Table.
(PDF)

**S6 Fig. Heatmap of sequence diversity in TprK V7.** Deep sequencing of TprK V7 region showing persistence of inoculum variants and generation of non-inoculum variants in samples collected overtime from rabbits infected with the WT SS14 strain ("WT"-labeled samples, left side of the map) and form rabbits infected with the SS14-DC$^{KO}$ strain ("KO"-labeled samples, right side of the map), separated by a bold vertical dashed line. In each map, inoculum sequences for the WT strain are labeled as "WT Inoc B00 Day 00", where WT Inoc indicates WT inoculum treponemes, B00 indicates that the sample was not obtained from a biopsy, and Day 00 indicates the experiment's time 0. The same nomenclature was adopted for the SS14-DC$^{KO}$ inoculum, with the exception that KO replaced WT. Samples collected post-inoculation report rabbit number (R#), biopsy number (B#), and day post-inoculation the sample was obtained (Day#). Light gray dashed lines separate individual animals. Missing samples did not yield data. The same heatmap in interactive format is available at https://github.com/greninger-lab/Impaired-TprK-Antigenic-Variation. Peptide sequences and prevalence are also reported in S2 Table.
(PDF)

**S1 File. Sequence of the insert (yellow/green/gray bases) contained in the p*DC*arms-47p-*kan*$^R$ suicide vector built using a pUC57 plasmid backbone, flanking regions (aqua), and primers used to evaluate the replacement of the DCs in the transformed *T. pallidum* strain.**
(DOCX)

**S1 Table. TprK variability profile in the WT SS14 and SS14-DS$^{KO}$ strains used for rabbit inoculation.** Highly represented sequences (>10% of the total) are in bold.
(DOCX)

**S2 Table. Results of the TprK profiling showing variant sequences with ≥0.25% prevalence for each V region, time point, and infected rabbit.**
(CSV)

**S3 Table. Inoculum variants detected in rabbits infected with the SS14-DC$^{KO}$ strain at <0.25% reads.** KO = rabbit infected with the DC$^{KO}$ strain; R = rabbit#; B = biopsy#; Day# = day post-inoculation; (%) = percentage of sequence compared to all sequences detected in the sample for a given V region at this time point.
(DOCX)

**S4 Table. Barcoded primers used in this study for *tprk* deep sequencing.**
(XLSX)

## Acknowledgments

The content of this study is solely the responsibility of the authors and does not necessarily represent the official views of the National Institutes of Health or Open Philanthropy.

## Author Contributions

**Conceptualization:** Nicole A. P. Lieberman, Alexander L. Greninger, Lorenzo Giacani.

**Data curation:** Nicole A. P. Lieberman, Barbara Molini, Lorenzo Giacani.

**Formal analysis:** Nicole A. P. Lieberman.

**Funding acquisition:** Lorenzo Giacani.

**Investigation:** Emily Romeis, Nicole A. P. Lieberman, Barbara Molini, Lauren C. Tantalo, Benjamin Chung, Quynh Phung, Carlos Avendaño, Anastassia Vorobieva, Alexander L. Greninger.

**Methodology:** Emily Romeis, Nicole A. P. Lieberman, Barbara Molini, Lauren C. Tantalo, Benjamin Chung, Quynh Phung, Carlos Avendaño, Anastassia Vorobieva, Alexander L. Greninger.

**Project administration:** Lorenzo Giacani.

**Writing – original draft:** Nicole A. P. Lieberman, Alexander L. Greninger, Lorenzo Giacani.

**Writing – review & editing:** Lorenzo Giacani.

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
