## [Decision Letter · Decision Letter 0]

25 Jan 2023

Lorenzo,

Thank you very much for submitting your manuscript "Treponema pallidum subsp. pallidum with an Artificially Impaired TprK Antigenic Variation System is Attenuated in the Rabbit Model of Syphilis" for consideration at PLOS Pathogens. Your manuscript was reviewed by members of the editorial board and by several independent reviewers. In light of the reviews (below), we would like to invite the resubmission of a revised version that takes into account the reviewers' comments. Although molecular genetic experiments are in their infancy in T. pallidum and complementation is not yet feasible, all three reviewers are concerned about off-target effects of your mutagenesis, so either generate and assay a second, independent mutant or thoroughly analyze all genomic differences between the wild-type parent and the mutant (as mentioned by Reviewer 1) as well as evaluate expression of genes flanking the genetic lesion (as noted by Reviewer 3). In addition, address all of the minor comments from the reviewers (particularly the extensive points raised by Reviewer 2).

We cannot make any decision about publication until we have seen the revised manuscript and your response to the reviewers' comments. Your revised manuscript is also likely to be sent to reviewers for further evaluation.

Best wishes,

Scott

D. Scott Samuels

Academic Editor

PLOS Pathogens

David Skurnik

Section Editor

PLOS Pathogens

Kasturi Haldar

Editor-in-Chief

PLOS Pathogens

orcid.org/0000-0001-5065-158X

Michael Malim

Editor-in-Chief

PLOS Pathogens

orcid.org/0000-0002-7699-2064

Reviewer's Responses to Questions

**Part I - Summary**

Reviewer #1: In their submitted manuscript, Giancani and colleagues build on their recently developed technological advance – genetic manipulation of Treponema pallidum. In prior studies, this same group described allelic replacement of the tprA locus with a kanamycin resistance marker using a suicide vector. This advance was made possible by the recent development of a system for cultivation of T pallidum in vitro by Norris and colleagues. In their first studies, the Giacani group opted to start with tprA because the corresponding gene product is non-essential. In the current manuscript, the authors extend their prior studies by attempting to knock out tprK in the reference strain SS14. Numerous prior studies by Giacani and others suggest that antigenic variation by TprK promotes immune evasion during natural infection. Although unsuccessful in knocking out tprK, the authors were able to eliminate most of. The cassettes used by tprK for antigenic switching. The resulting SS14-DC_ko strain grew normally in rabbit testes and could be sustained through in vitro passaging but was unable to produce cutaneous lesions on the backs of rabbits inoculated intradermally. Rabbits infected with the DC_ko also failed to seroconvert, further highlighting the importance of TprK antigenic switching for persistence.

Overall, the manuscript is well written, and the authors conclusions are largely supported by the data. The development of a facile system for generating mutants in T pallidum is a major advance for the field in general. The workflow used by Giacani and colleagues to characterize the DC-ko mutant is very through, as would be expected of this group. While the work presented in the submitted manuscript are a major accomplishment. there are some concerns, outlined below, that need to be addressed.

Reviewer #2: This manuscript by Romeis et al. provides evidence for the importance of tprK sequence variation in the ability of Treponema pallidum to cause persistent infection in mammalian hosts. Deletion of a major portion of the donor sites for tprK variation led to reduced T. pallidum burden, lesion development, and tprK sequence variation in a rabbit model. Importantly, this represents for the first report of a functional mutation in T. pallidum. Limitations include the use of a heterogeneous strain as the starting point and the lack of complementation and thus fulfillment of ‘molecular Koch’s postulates’; the manuscript should be modified to further recognize these aspects. Nevertheless, the manuscript represents a significant step in the study of T. pallidum pathogenesis.

Reviewer #3: The present study by Romais et al sets out to capitalize on a recently developed tool to genetically manipulate the causative agent of syphilis, T. pallidum. Utilizing this tool, the authors successfully deleted 96% of the donor cassettes that provide variation to the tprK gene. Their results demonstrate that the genetically altered T. pallidum SS14 strain is impaired in its ability to vary the TprK antigen, and exhibits attenuated proliferation and disease during infection of rabbits compared to the wild type strain. This work represents the first of its kind on TprK antigenic variation and has the potential to significantly advance our knowledge on the importance of TprK for T. pallidum virulence and persistence. However, the study suffers from a few key weaknesses, most notably the absence of any attempt to complement the observed phenotypes.

**Part II – Major Issues: Key Experiments Required for Acceptance**

Reviewer #1: TprK variants in Table 3 indicate that the wildtype and DC-KO are non-clonal. Did the authors isolate only a single DC_ko isolate? Were attempts made to perform limiting dilution to get clonal isolates? The authors indicate that the genomes of the strains used for infection studies were sequenced, but only provide data for tprK. In the absence of complementation. A more throughout analysis of SNPs between the two strains used for IT and ID inoculation is warranted.

Reviewer #2: (No Response)

Reviewer #3: Although the authors have done a commendable job validating the SS14-DCKO mutant clone that includes confirming the singular presence of the integrated kanR gene, the study suffers from the absence of a complement clone for use in infection assays. Understandably, genetic manipulation is only just coming of age for Tp, but the need for some type of complementation to ensure that all mutation-related phenotypes are solely due to loss DCs is especially important given that this is the first time this region has been genetically manipulated. In lieu of a genetic complement, the RIF experiments should at least be carried out in immune-deficient or immune-suppressed rabbits to verify that they behave similar to WT in the absence of a host antibody response. Propagation of the mutant in the immune-privileged testis is not sufficient because it does not involve the same infection processes as those in the RIF experiments.

In line with the above comment, have the authors looked into the possibility of polar effects on genes flanking the deleted DC regions? Again, this seems warranted to ensure that all observed mutant phenotypes are due to absence of the deleted DCs.

Pgs. 12-13, lines 244-246 & 277-280 – The detection of Tp DNA via qPCR appears to contradict the treponemal clearance observed with the RIT. This would seem to suggest that the qPCR only detected residual gDNA from dead treponeme cells. Did the authors attempt to culture Tp recovered from the biopsies and aspirates to determine viability?

**Part III – Minor Issues: Editorial and Data Presentation Modifications**

Reviewer #1: The authors noted a difference in lesion development between wt and DC_ko isolates as early as 9 days post-inoculation, well before detectable antibody titers for TprK were observed. Moreover, the authors noted the relatively low levels of expression for tprK. These data seem to argue against Ab-mediated clearance as being responsible for this difference.

Reviewer #2: 1. It would be useful to present any information regarding the transformation frequency. Were the number of ‘input’ bacteria (i.e. the number of T. pallidum adherent at the time of CaCl + DNA treatment) determined? Were all of the wells consistently positive for outgrowth of KanR T. pallidum, or only a portion (e.g. 2/24 wells) positive? Can the frequency of transformation be estimated through a combination of these data and the time of growth detection?

2. Fig. 5 and accompanying text. qPCR determinations in these kinds of studies are typically normalized to amplifications of a host gene, whereas in this case are presented as “tp0574 copies/reaction”. Ref. 56, which is cited for the procedure, utilized the human beta-globin gene for normalization. The M&M section does not provide any additional information, such as the amount of DNA used. The amount of DNA purified from each sample would be expected to vary widely, based on size of the biopsy, mincing, efficacy of extraction, etc., so some method for normalization is needed.

3. l. 268. Was qPCR performed on the popliteal lymph node samples as another measure of T. pallidum quantity?

4. l. 322-324. This statement is unclear as written.

5. The labeling in Fig. 6 is far too small to visualize in the journal format. Also, in this and in similar supplemental figures, the SS14-WT and SS14-KO regions are not labeled. Another way to present this information in the article should be developed; each of the V region datasets could then be included as supplemental figures.

6. Table 3. For regions V5 and V6, the predominant genotypes for SS14-WT and SS14-KO. This pattern is also evident in the T. pallidum recovered from infected rabbits (e.g. Fig. 6). A possible explanation is that a subset of clones in the SS14-WT population were mutated. Possible explanations should be provided in the Discussion.

7. l. 373-376. This statement represents part of the results, and should be moved to the Results section.

8. l. 395-398. Only the seminiferous tubules of the testes and associated ducts represent an immune-privileged site. The connective tissue where T. pallidum propagates is not immune-protective, as indicated by the rapid elimination of the bacteria after infiltration of the site by T cells and macrophages. The particular time point that the authors examined apparently was before this infiltration occurred. It is likely that a similar pattern of more rapid elimination of SS14-KO from the testes would have been evident if a time course were performed. The text should be modified to reflect this consideration.

9. l. 420. Most would argue that not beginning with an isogenic population and isolating the KO mutant from that strain is a true limitation in this study; it complicates the interpretation of the data, as exemplified by tprK sequence variation results. Starting with a clonal population is the paradigm for bacterial genetic studies. It may be best to accept this limitation and state that more definitive studies beginning with a clonal population will be done in the future.

10. Another limitation is the lack of complementation and restoration of WT infectivity patterns. Thus it is possible that the observed phenotype is due to other ‘background’ mutations in the recovered SS14-KO strain. This is implied, but should be stated more clearly in the Discussion.

11. l.430-443. Interpretation of these results is complicated by the lack of long-term serologic testing in RIT recipients of lymph node extracts from 505 and 512 (Table 2). Also, both recipients receiving the day 79 inoculum seroconverted, contrary to line 432. It is suggested that this section be deleted or modified.

12. l. 449. The wording “led to clearance” is an overstatement, in that T cell responses were likely also important. Perhaps ‘likely contributed to clearance’?

13. l. 465. The ‘unable to proliferate’ statement seems contradictory to l. 160-164, and should be modified.

14. l. 484. Please indicate whether the rabbits were kept in a low-temperature facility and provided antibiotic-free food and water.

15. None of the supplemental figures or tables have titles, legends or footnotes (except for Table S3). Please provide this information.

Minor

l. 164. Please provide the total number of T. pallidum extracted, rather than T. pallidum/ml (which the reader can’t interpret without knowing the number of mls extracted).

Figure legends. Each figure legend should start with a title encompassing the overall figure (before A).

l. 214-216. Change figure references to Fig. 3F and Fig. 3G.

Fig. 4. The statistical ‘brackets’ used are confusing because they resemble the bars, and should be replaced by a single horizontal line. Also, the “ns” designation for the last timepoints is not referred to in either the legend or the text; it does not seem to be important, and should be removed from the figure.

l. 546-547. old medium … fresh medium.

l. 698. Indicate the size of the punch biopsies.

l. 708. “in case” is incomplete.

Reviewer #3: Pg. 16, lines 324-325 – This statement seems a bit overreaching given that a number of variants are detected in the DC KO mutant. Something to the extent of “significantly decreased” or similar seems more in line with the results.

Pg. 20, lines 412-414 & 428-430 – It seems unlikely that a highly conserved DC in Tp would play a marginal role in TprK antigenic variation. Other possibilities are more plausible, including that V7 variation may be more important during late-stage infection. Such alternatives should be at least briefly mentioned in the text.

Pg. 23, line 501 – text should be revised to read “in a previous study”.

Pg. 38, line 838 – revise “form” to “from”.

Throughout the manuscript – need more consistency with the use of WT treponemes vs wild-type treponemes.

PLOS authors have the option to publish the peer review history of their article (what does this mean?). If published, this will include your full peer review and any attached files.

Reviewer #1: No

Reviewer #2: No

Reviewer #3: No
---

## [Editor Report · Decision Letter 1]

1 Mar 2023

Lorenzo,

We are pleased to inform you that your manuscript 'Treponema pallidum subsp. pallidum with an Artificially Impaired TprK Antigenic Variation System is Attenuated in the Rabbit Model of Syphilis' has been provisionally accepted for publication in PLOS Pathogens.

Best wishes,

Scott

D. Scott Samuels

Academic Editor

PLOS Pathogens

David Skurnik

Section Editor

PLOS Pathogens

Kasturi Haldar

Editor-in-Chief

PLOS Pathogens

orcid.org/0000-0001-5065-158X

Michael Malim

Editor-in-Chief

PLOS Pathogens

orcid.org/0000-0002-7699-2064
---

## [Editor Report · Acceptance letter]

13 Mar 2023

Dear Dr. Giacani,

We are delighted to inform you that your manuscript, "Treponema pallidum subsp. pallidum with an Artificially Impaired TprK Antigenic Variation System is Attenuated in the Rabbit Model of Syphilis," has been formally accepted for publication in PLOS Pathogens.

Best regards,

Kasturi Haldar

Editor-in-Chief

PLOS Pathogens

orcid.org/0000-0001-5065-158X

Michael Malim

Editor-in-Chief

PLOS Pathogens

orcid.org/0000-0002-7699-2064